# TABSWIFT: An Efficient Tabular Foundation Model with Row-Wise Attention

**Si-Yang Liu** [1 2]  **Han-Jia Ye** [1 2]

## Abstract

Tabular foundation models, exemplified by TabPFN, perform prediction via in-context learning, inferring test labels directly from labeled training examples. They have demonstrated competitive performance, particularly on small-to-medium datasets. However, recent tabular foundation models often improve accuracy with increasingly complex architectures, incurring higher inference cost and limiting practical deployment. In this work, we revisit the original TabPFN design and show that a lightweight row-wise attention–only backbone can remain highly competitive with two simple enhancements: a gated attention stabilization mechanism and a small set of learnable register tokens that provide global context and improve pretraining quality. The resulting model, TABSWIFT, supports both classification and regression, and is competitive with stronger tabular foundation models (e.g., TabPFN v2 and TabICL) while being more efficient at inference. For latency-sensitive serving, we further introduce an adaptive layer-wise early-exit mechanism that dynamically adjusts inference depth per sample. Overall, TABSWIFT enables efficient and anytime tabular in-context learning for practical deployments.

## 1. Introduction

Tabular data is ubiquitous across a wide range of real-world applications, including healthcare (Hassan et al., 2020), finance (Cao & Tay, 2001), e-commerce (Nederstigt et al., 2014), and scientific research (Tibshirani et al., 2002). In this format, each instance is represented as a vector of heterogeneous attributes, and the goal is to map these feature vectors to discrete labels or continuous targets. Despite

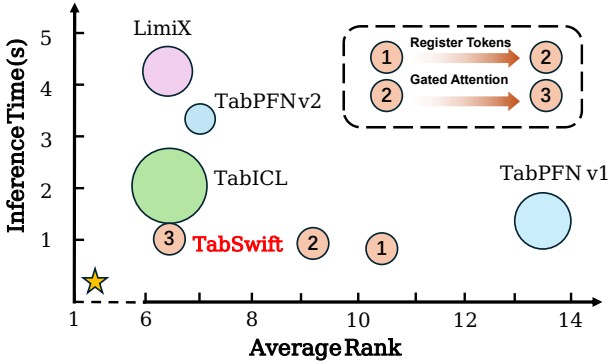

*Figure 1.* **Performance–efficiency–size** comparison on the TAL-ENT benchmark. Each model is plotted by its *average rank* (lower is better, x-axis) and *average inference time* (lower is better, y-axis); bubble area encodes model size. Following the convention of TALENT (Ye et al., 2024), the ⋆ marks the *ideal point*—a hypothetical model with both the best predictive performance and the lowest inference latency. TABSWIFT (ours) is the orange marker labeled "3". TABSWIFT attains an average rank comparable to other TFMs but at substantially lower inference cost, illustrating a markedly better accuracy–efficiency trade-off.

the success of deep learning in vision and language, predictive modeling on tabular data has long been dominated by tree-based methods, especially gradient-boosted decision trees (GBDTs) (Chen & Guestrin, 2016; Ke et al., 2017), due to their strong accuracy, robustness to feature scaling and missing values, and reliable performance on small datasets (McElfresh et al., 2023; Ye et al., 2024).

Tabular learning has resisted a foundation-model-style breakthrough largely due to its heterogeneity and distribution diversity (Jiang et al., 2025). Feature dimensionalities and semantics vary substantially across datasets (mixed numerical/categorical types, missing values, and noisy attributes) (Borisov et al., 2024), and unlike language or vision, tables lack a universal vocabulary or spatial structure to support a transferable tokenization. This difficulty is exacerbated by the small size of many real-world datasets, where training from scratch is brittle and hyperparameter-sensitive, motivating tabular predictors that transfer *out of the box* with minimal tuning (van Breugel & van der Schaar, 2024).

Recently, tabular foundation models have largely followed the Prior-Fitted Network (PFN) paradigm and perform prediction via *in-context learning* (ICL): a pretrained trans-

---

[1]School of Artificial Intelligence, Nanjing University, China [2]National Key Laboratory for Novel Software Technology, Nanjing University, China. Correspondence to: Han-Jia Ye <yehj@lamda.nju.edu.cn>.

*Proceedings of the 43rd International Conference on Machine Learning*, Seoul, South Korea. PMLR 306, 2026. Copyright 2026 by the author(s).

former maps a labeled support set and a query instance to a prediction without test-time parameter updates (Hollmann et al., 2023; Brown et al., 2020). TabPFN demonstrated that this amortized inference approach can yield strong performance on tabular classification, especially in small-data regimes, and TabPFN v2 (Hollmann et al., 2025) further extends PFNs to both classification and regression with improved performance across a range of settings.

From an architectural perspective, existing ICL-based tabular foundation models can be broadly categorized into two families. The first family follows the original TabPFN v1 design and relies on *row-wise attention–only* (e.g., TabPFN, TabForestPFN (den Breejen et al., 2024), and TabDPT (Ma et al., 2025)): instances are embedded as row tokens and self-attention is applied only across rows. This minimalist backbone generally keeps inference cost moderate compared to more complex designs. However, relying solely on row-wise mixing may limit the model's capacity to capture richer *feature-wise* structure, which can translate into a lower performance ceiling on highly heterogeneous tasks.

The second family introduces explicit *feature-wise* modeling, for example by incorporating alternating row/column attention patterns or additional embedding modules. TabPFN v2 and several follow-up works (e.g., TabICL (Qu et al., 2025) and LimiX (Zhang et al., 2025)) adopt such components to better capture feature-wise structure and dataset-specific heterogeneity. While effective, these designs substantially increase inference cost compared to the TabPFN v1 row-wise attention–only backbone: alternating row/column attention introduces extra attention passes over features and rows, making each forward pass and the overall inference pipeline more compute- and memory-intensive than the minimalist TabPFN v1 design.

In many practical tabular applications, prediction is performed under strict inference budgets (e.g., per-query latency and throughput constraints), making the accuracy–efficiency trade-off a first-order concern. Motivated by the two prevailing architectural routes discussed above—lightweight row-wise attention-only tabular foundation models versus more expressive but heavier column-aware designs—we revisit a simple yet underexplored question:

> *Can the row-wise attention-only backbone of TFMs achieve competitive foundation-model-style performance when trained with modern attention stabilization and pretraining techniques?*

We answer this question affirmatively and propose TAB-SWIFT, a lightweight tabular foundation model that retains the row-wise attention-only inference structure while incorporating two simple enhancements: element-wise *gated attention* for stabilizing pretraining and a small set of learnable *register tokens* as task-shared latent slots to aggregate global context. Despite its streamlined attention structure, TAB-

SWIFT attains performance competitive with more complex tabular foundation models while offering a more favorable efficiency profile, as shown in Figure 1. Moreover, we train TABSWIFT under a unified formulation that supports *both classification and regression*, enabling a single pretrained model to serve a broader range of downstream tabular tasks.

Beyond accelerating the backbone architecture, we further address a deployment-centric challenge: even an efficient transformer can be wasteful if every test instance must traverse all layers. This is especially relevant for per-sample inference, where latency is dominated by the worst-case computation depth. To this end, we introduce an *adaptive layer-wise early-exit* mechanism on top of the pretrained TABSWIFT. Specifically, we attach an auxiliary prediction head to each transformer layer so that intermediate representations can produce valid predictions, and we learn an additional gating head that predicts whether the current layer's prediction is reliable enough to stop and exits as soon as it is deemed reliable. Empirically, we observe that the majority of test instances can be confidently predicted at shallow depths, while only a small fraction requires full-depth computation. This yields an *anytime* tabular foundation model that adapts its computation to the difficulty of each input.

Our main contributions are as follows:

- We propose TABSWIFT, showing that a row-wise attention–only PFN backbone can remain highly competitive when stabilized by *gated attention* and augmented with learnable *register tokens*, providing a strong accuracy–efficiency trade-off. The code is available at https://github.com/LAMDA-Tabular/TabSwift.
- We develop a unified tabular in-context pretraining formulation that supports both classification and regression within a single pretrained model.
- We introduce a reliability-predictive adaptive layer-wise early-exit strategy for per-sample deployment, enabling anytime inference and substantially reducing average computation with minimal loss in predictive performance.

## 2. Preliminaries

**Tabular prediction tasks.** A tabular dataset is a collection of $N$ instances with $d$ heterogeneous attributes (columns). For supervised learning, we observe $\mathcal{D} = \{(\mathbf{x}_i, y_i)\}_{i=1}^{N}$ with $\mathbf{x}_i \in \mathbb{R}^d$ (after categorical encoding) and targets $y_i$ for either classification ($y_i \in \{1, \ldots, C\}$) or regression ($y_i \in \mathbb{R}$). The goal is to learn a predictor $f_\theta : \mathbb{R}^d \to \mathcal{Y}$ that minimizes the expected risk $\mathbb{E}_{(\mathbf{x},y)\sim\mathcal{P}}[\ell(f_\theta(\mathbf{x}), y)]$.

**Tabular foundation models and in-context learning.** A tabular foundation model aims to learn parameters that generalize across a distribution of tasks. At inference time, the model is conditioned on a *context* (support set) and predicts targets for new queries without task-specific gradient up-

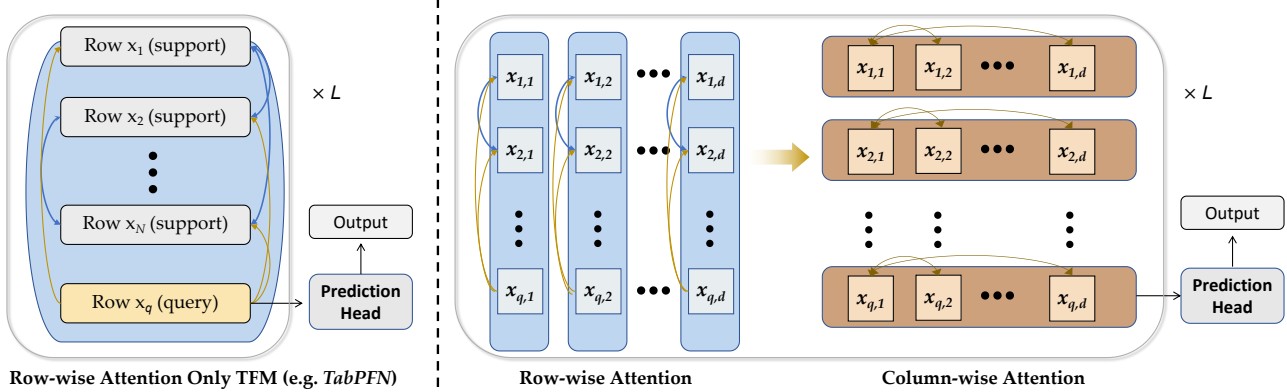

*Figure 2.* **Attention patterns for tabular in-context learning. Left:** Row-wise attention–only backbones (e.g., TabPFN) treat each row as a token and apply self-attention across the $n$ in-context rows (support rows plus the query), followed by a prediction head on the query row. **Right:** Alternating row/column attention views the input as an $n \times d$ grid and alternates (i) row-wise mixing across instances for each attribute and (ii) column-wise mixing across attributes for each instance, typically yielding higher per-layer cost.

dates, i.e., it approximates a conditional predictor of the form $f_\theta(\mathbf{x}_q; \mathcal{C}_\tau)$. In PFN-style tabular ICL, the context is a labeled support set $\mathcal{D}_{\text{sup}} = \{(\mathbf{x}_i, y_i)\}_{i=1}^N$ and prediction is performed by feeding $(\mathcal{D}_{\text{sup}}, \mathbf{x}_q)$ as a prompt to a pretrained transformer.

**Attention cost.** We consider a labeled *support* set $\mathcal{D}_{\text{sup}} = \{(\mathbf{x}_i, y_i)\}_{i=1}^N$ and an unlabeled *query* instance $\mathbf{x}_q$. Let $d$ denote the number of attributes (columns) and let $n = N + 1$ denote the number of rows processed in-context (support rows plus the query row). We use the standard multi-head scaled dot-product attention

$$\text{Attn}(\mathbf{Q}, \mathbf{K}, \mathbf{V}) = \text{softmax}\left(\frac{\mathbf{Q}\mathbf{K}^\top}{\sqrt{d_h}}\right)\mathbf{V}, \qquad (1)$$

where $d_h$ is the per-head dimension and $d_{\text{model}} = h\, d_h$ for $h$ attention heads. Unless otherwise stated, we focus on the quadratic attention term whose dominant per-layer cost is

$$\mathcal{O}\big(h S^2 d_h\big) = \mathcal{O}\big(S^2 d_{\text{model}}\big), \qquad (2)$$

for sequence length $S$, omitting linear projection and MLP terms.

**PFN with row-wise attention.** Prior-Fitted Networks (PFNs) pretrain a transformer over a distribution of synthetic tabular tasks and perform prediction via in-context learning (ICL), i.e., producing $\hat{y}_q = f_\theta(\mathcal{D}_{\text{sup}}, \mathbf{x}_q)$ without test-time parameter updates. In the original TabPFN-style backbone, instances are embedded into *row tokens* and self-attention is applied only across rows (i.e., across the $n$ in-context instances). Concretely, given a query instance $\mathbf{x}_q$ and a labeled support set $\mathcal{D}_{\text{sup}}$ of size $N$, we build a prompt of $n = N + 1$ rows and form row representations $\mathbf{H}^{(0)} \in \mathbb{R}^{n \times d_{\text{model}}}$ from $(\mathbf{x}_i, y_i)$ for $i \leq N$ and from $\mathbf{x}_q$ for the query. We then apply $L$ transformer layers

$$\mathbf{H}^{(\ell+1)} = \text{TF}\big(\mathbf{H}^{(\ell)}\big), \quad \ell = 0, \dots, L-1, \qquad (3)$$

where $\text{TF}(\cdot)$ uses row-wise self-attention over the $n$ prompt tokens, and a prediction head reads out the query row to output $\hat{y}_q$. From the perspective of *per-query* inference, the dominant per-layer attention cost is $\mathcal{O}(n^2 d_{\text{model}})$, yielding total cost $\mathcal{O}(Ln^2 d_{\text{model}})$. This row-wise attention-only design is computationally attractive, but more recent PFN variants often add additional structure to better handle heterogeneity (Ye et al., 2025a).

**Alternating row/column attention.** To explicitly model both *instance-wise* and *attribute-wise* interactions, several tabular foundation models employ alternating attention patterns, e.g., applying row attention and column attention in sequence (Hollmann et al., 2025). As shown in Figure 2, a convenient abstraction is to view the input as an $n \times d$ grid of tokens indexed by row and column, and alternate: (i) *row attention*, which applies attention over the $n$ rows *for each* of the $d$ attributes, and (ii) *column attention*, which applies attention over the $d$ attributes *for each* of the $n$ rows. Ignoring constant factors, one alternating block has cost

$$\mathcal{O}\big(d\, n^2 d_{\text{model}}\big) \;+\; \mathcal{O}\big(n\, d^2 d_{\text{model}}\big), \qquad (4)$$

capturing the dominant quadratic attention terms when attention is applied along one axis at a time. While effective, alternating patterns and extra embedding/tokenization modules generally make a single forward pass heavier than a minimalist row-wise attention–only backbone, motivating the accuracy–efficiency trade-offs discussed in Section 1. A detailed discussion of related work is provided in Appendix A

## 3. TABSWIFT

We present TABSWIFT, a lightweight tabular foundation model designed for efficient in-context learning (ICL). Our method builds on the row-attention-only design of

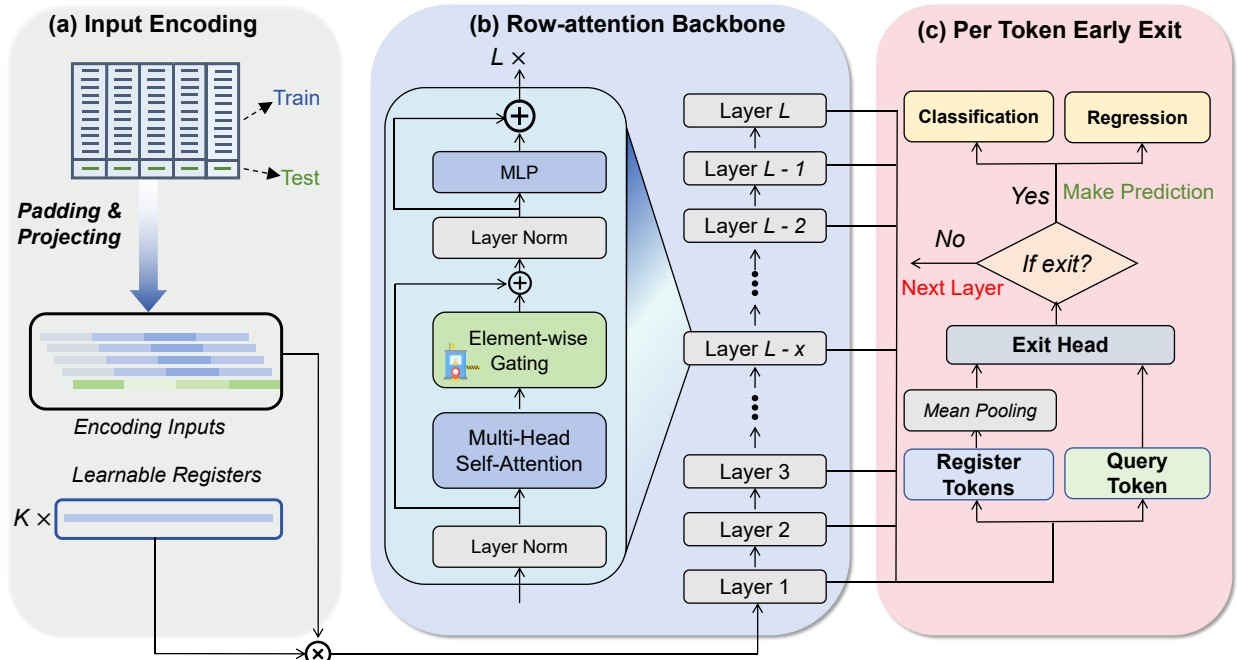

*Figure 3.* **TABSWIFT overview.** (a) Input encoding standardizes each task to a fixed feature dimension $F_{\max}$ (zero-padding if $F < F_{\max}$; PCA projection if $F > F_{\max}$), embeds each row as a token, and prepends $K$ learnable register tokens. (b) A row-attention-only transformer backbone processes the token sequence, equipped with element-wise SDPA-output gated attention (G1) to modulate attention updates. (c) Adaptive per-token early exit attaches prediction heads and register-conditioned exit heads to the last $E$ layers; the exit head takes $[\mathbf{z}^{(e)}; \mathrm{mean}(\mathbf{R}^{(e)})]$ and selects the earliest exit satisfying $\sigma(s^{(e)}) \geq \tau$, yielding an accuracy–efficiency trade-off.

TabPFN (Hollmann et al., 2023), and introduces three key enhancements: (1) a stabilized transformer backbone with register tokens and gated attention; (2) an adaptive early-exit mechanism for per-sample inference-time efficiency; and (3) a unified pretraining setup that supports both classification and regression tasks. This section describes each component in detail.

### 3.1. Model Architecture

TABSWIFT is a lightweight PFN-style transformer for tabular in-context learning (ICL). It follows the row-attention-only backbone of TabPFN (Hollmann et al., 2023)—treating each table row as a token—and introduces two lightweight architectural modifications: (i) a small set of learnable *register tokens* and (ii) an element-wise *gated attention* variant in the self-attention blocks.

**Row tokenization and input encoding.** Given a labeled support set $\mathcal{D}_{\mathrm{sup}} = \{(\mathbf{x}_i, y_i)\}_{i=1}^N$ and an unlabeled query instance $\mathbf{x}_q$, we construct a prompt consisting of $n = N + 1$ rows. To handle varying feature dimensions across datasets, we standardize each input to a fixed feature dimension $F_{\max}$: for a table with $F$ features, we zero-pad if $F < F_{\max}$ and

apply PCA if $F > F_{\max}$, yielding $\tilde{\mathbf{x}} \in \mathbb{R}^{F_{\max}}$.[1] We then normalize each row independently (e.g., per-row z-score on numerical features after basic preprocessing) and embed it into a $d_{\mathrm{model}}$-dimensional row token:

$$\mathbf{e}(\mathbf{x}) = \phi(\tilde{\mathbf{x}})\mathbf{W}_x + \mathbf{b}_x \in \mathbb{R}^{d_{\mathrm{model}}}, \qquad (5)$$

where $\phi(\cdot)$ denotes the normalized feature vector (and any fixed encoding for categorical values, if used). For support rows, we additionally embed the label and inject it into the token:

$$\mathbf{h}_i^{(0)} = \mathbf{e}(\mathbf{x}_i) + \mathbf{e}_y(y_i), \ i \leq N; \ \ \mathbf{h}_q^{(0)} = \mathbf{e}(\mathbf{x}_q), \quad (6)$$

producing an initial row-token matrix $\mathbf{H}_{\mathrm{rows}}^{(0)} \in \mathbb{R}^{n \times d_{\mathrm{model}}}$. Following the practice of augmenting Transformer inputs with a small number of learnable global tokens to store and propagate task-level information, we prepend $K$ learnable *register tokens* $\mathbf{R}^{(0)} = \{\mathbf{r}_k^{(0)}\}_{k=1}^K$ (Darcet et al., 2024). These tokens are shared across tasks, concatenated to the

---

[1] For datasets with very large feature dimensions, the PCA preprocessing step introduces additional computational overhead. Therefore, the runtime advantage of TABSWIFT in such cases should be understood primarily as coming from its lightweight row-attention-only design and smaller constant factors, rather than from a strictly better asymptotic complexity in all regimes.

row-token sequence, and updated through all layers. Such global tokens can act as latent slots that aggregate contextual information and make it easier for the model to maintain and refine dataset-level representations across depth (Darcet et al., 2024). The layer input is thus

$$\mathbf{H}^{(0)} = \left[\mathbf{R}^{(0)}; \mathbf{H}^{(0)}_{\text{rows}}\right] \in \mathbb{R}^{S \times d_{\text{model}}}, \quad S = K + n. \quad (7)$$

We note that recent PFN variants (Grinsztajn et al., 2025; Bouadi et al., 2025) employ a similar mechanism by introducing a set of task-level latent tokens, which they refer to as "thinking rows".

**Row-wise attention-only transformer with element-wise gated attention.** We apply $L$ transformer layers to $\mathbf{H}^{(0)}$. Each layer performs multi-head self-attention (MHSA) *over the token sequence* (i.e., across the $S$ tokens), followed by a position-wise MLP. Let $\mathbf{X} \in \mathbb{R}^{S \times d_{\text{model}}}$ denote the layer input (we omit the layer index $\ell$ for clarity) and $h \in \{1, \ldots, H\}$ index heads with per-head dimension $d_h$. Standard attention computes

$$\mathbf{Q}^h = \mathbf{X}\mathbf{W}^h_Q, \ \mathbf{K}^h = \mathbf{X}\mathbf{W}^h_K, \ \mathbf{V}^h = \mathbf{X}\mathbf{W}^h_V,$$
$$\mathbf{A}^h = \text{softmax}\left(\frac{\mathbf{Q}^h(\mathbf{K}^h)^\top}{\sqrt{d_h}}\right), \ \mathbf{O}^h = \mathbf{A}^h\mathbf{V}^h. \quad (8)$$

We adopt *SDPA-output gating* (the G1 design in (Qiu et al., 2025)), which applies an element-wise multiplicative gate to the SDPA output before the output projection:

$$\mathbf{G}^h = \sigma(\mathbf{X}\mathbf{W}^h_g + \mathbf{b}^h_g) \in (0,1)^{S \times d_h}, \ \widetilde{\mathbf{O}}^h = \mathbf{G}^h \odot \mathbf{O}^h. \quad (9)$$

The gated MHSA output is

$$\text{G-MHSA}(\mathbf{X}) = \text{Concat}_h(\widetilde{\mathbf{O}}^h)\mathbf{W}_O. \quad (10)$$

Recent analysis of gated attention in LLMs suggests that such output gating can introduce an additional non-linearity between the value and output projections and may induce input-dependent sparsity in attention updates (Qiu et al., 2025). Motivated by these findings, we use gated attention as a lightweight modification to improve optimization stability at negligible architectural overhead in our synthetic pretraining setup.

With pre-norm and residual connections, layer $\ell$ is implemented as

$$\mathbf{U}^{(\ell)} = \mathbf{H}^{(\ell)} + \text{G-MHSA}(\text{LN}(\mathbf{H}^{(\ell)})), \quad (11)$$
$$\mathbf{H}^{(\ell+1)} = \mathbf{U}^{(\ell)} + \text{MLP}(\text{LN}(\mathbf{U}^{(\ell)})). \quad (12)$$

**Prediction heads.** After $L$ layers, we read out the query-row representation $\mathbf{h}^{(L)}_q$ (the token corresponding to $\mathbf{x}_q$) and feed it into a task-type-specific head to obtain $\hat{y}_q$. We use separate lightweight MLP heads for classification and regression (sharing the same backbone), enabling one pretrained model to serve both task types.

**Complexity.** Let $S = n + K$ be the token count. The dominant attention cost per layer is $\mathcal{O}(S^2 d_{\text{model}})$, so the backbone cost is $\mathcal{O}(LS^2 d_{\text{model}})$. The gating computation adds a linear term $\mathcal{O}(S d_{\text{model}})$ per layer, which is negligible compared to the quadratic attention term for typical $S$. Overall, the row-wise attention-only design remains substantially cheaper than architectures that additionally perform attribute-wise (column) attention.

### 3.2. Adaptive Early-Exit for Tabular ICL

Building on the row-attention-only TABSWIFT backbone introduced above, we further equip the model with an adaptive early-exit mechanism to *reduce average inference cost* by allocating less computation to "easy" queries and more to "hard" ones. Prior work has explored early exiting for tabular ICL by attaching intermediate decoders and using confidence signals to select an exit depth (Küken et al., 2025). A key distinction is the inference setting: some tabular ICL early-exit protocols rely on *test-set-level* statistics to determine the stopping depth, which assumes access to the test set as a whole and may not reflect strict online per-query deployment. In contrast, we target *per-query* inference, where each query arrives independently and the exit decision must be made for that query alone. Concretely, we attach lightweight *prediction heads* and learned *exit heads* to a subset of transformer layers, enabling intermediate predictions and an input-dependent stopping decision.

**Layer-wise exits.** Let $\mathbf{H}^{(\ell)} \in \mathbb{R}^{S \times d_{\text{model}}}$ denote the hidden states at layer $\ell$, where $S = n + K$ is the number of tokens (row tokens plus $K$ registers). We place exits on the last $E$ layers of the encoder (i.e., we use $E = L$ in our implementation). Let $T$ denote the number of query/test positions decoded in one forward pass (typically $T = 1$ for per-query inference). For each exit layer $e \in \{1, \ldots, E\}$, we apply a task-type-specific prediction head to the hidden states of query/test positions. Denote by $\mathbf{z}^{(e)} \in \mathbb{R}^{T \times d_{\text{model}}}$ the corresponding hidden states at layer $e$. The prediction head outputs

$$\hat{\mathbf{y}}^{(e)} = h^{(e)}_{\text{pred}}\left(\mathbf{z}^{(e)}\right), \quad (13)$$

where $h^{(e)}_{\text{pred}}(\cdot)$ is an MLP producing either classification logits (over up to $C_{\text{max}}$ classes) or regression values (scalar output). The last exit ($e = E$) corresponds to the full-depth prediction.

**Register-conditioned exit heads.** In addition to predicting labels, each exit layer produces a gating signal indicating whether the current-depth prediction is sufficiently reliable to stop. Our exit heads leverage both (i) the current query/test representation and (ii) a compact summary of the register tokens, which provide task-level context accumulated so far. Let $\mathbf{R}^{(e)} \in \mathbb{R}^{K \times d_{\text{model}}}$ denote the register-token hidden states at exit layer $e$. We form a summary vector by

average pooling:

$$\mathbf{r}^{(e)} = \frac{1}{K} \sum_{k=1}^{K} \mathbf{R}_k^{(e)} \ \in \ \mathbb{R}^{d_{\text{model}}}. \qquad (14)$$

For each query/test token representation $\mathbf{z}_t^{(e)} \in \mathbb{R}^{d_{\text{model}}}$, we concatenate it with the broadcasted register summary and apply a lightweight MLP exit head:

$$s_t^{(e)} = h_{\text{exit}}^{(e)}\Big(\big[\mathbf{z}_t^{(e)}; \ \mathbf{r}^{(e)}\big]\Big) \ \in \ \mathbb{R}, \qquad (15)$$

where $s_t^{(e)}$ is a logit, and we interpret $\sigma(s_t^{(e)})$ as a stopping score (larger indicates higher predicted reliability) under the training criterion defined in Appendix B.2.

**Inference-time stopping policy.** During inference, we process layers sequentially and evaluate exits from shallow to deep. For each query/test token $t$, we select the *earliest* exit whose exit score exceeds a threshold $\tau$:

$$e_t^\star = \min \Big\{ e \in \{1, \dots, E\} \ : \ \sigma\Big(s_t^{(e)}\Big) \geq \tau \Big\}, \qquad (16)$$

and output $\hat{\mathbf{y}}_t^{(e_t^\star)}$. If no exit satisfies the criterion, we fall back to the final exit $E$. The threshold $\tau$ controls the accuracy–efficiency trade-off and can be chosen on a validation set. This policy yields sample-wise dynamic depth and does not require access to other test instances. We note that **early exiting for tabular ICL** has been explored in prior work (Küken et al., 2025), which attaches intermediate decoders and uses entropy-based signals to determine an exit depth. However, their evaluation setting is often closer to *test-time adaptation* (Wang et al., 2021) and *transductive inference* (Joachims, 1999): the stopping depth can leverage *test-set-level* statistics computed over the test batch as a whole. This differs from latency-critical serving, where each query arrives independently and such global test-set information is unavailable. In contrast, our stopping policy targets strict per-query (online) inference, making an exit decision solely from the current query's intermediate representations (and the register summary) within a single forward pass, without access to other test instances.

## 4. Experiments

We evaluate TABSWIFT on a large-scale public tabular benchmark——TALENT following the protocols of Ye et al. and Qu et al., and additionally extend the evaluation to regression tasks. Our experiments aim to answer: (i) whether a well-trained row-wise-attention-only tabular foundation model can be competitive with stronger tabular foundation models, (ii) how each architectural component contributes, and (iii) whether per-sample early exit provides favorable accuracy–latency trade-offs.

### 4.1. Benchmarks and Evaluation Protocol

**Benchmark.** We follow the experimental setting used in prior work (Qu et al., 2025) on the TALENT benchmark. TALENT is a large public benchmark collected from diverse sources, containing 300 binary, multi-class classification and regression datasets (Liu et al., 2025). Following (Gorishniy et al., 2021; Ye et al., 2024; 2025b), each dataset is randomly split into train/validation/test with proportions of 64%, 16%, and 20%, respectively, and results are averaged over 15 random seeds. For classification, we report AUC (higher is better); For regression, we report RMSE (lower is better).

**Methods Compared.** We compare TABSWIFT with a diverse set of baselines spanning classical machine learning, tabular deep models, and recent tabular foundation models. For brevity, we defer the full list of compared methods and implementation details to the appendix (see Appendix B).

### 4.2. Training Details

**Pretraining data generation.** We follow the pretraining framework and synthetic data generation protocol of TabICL (Qu et al., 2025). Concretely, we *offline* generate a pool of 20,000 pretraining steps, where each step contains 512 independently sampled synthetic tabular tasks. For each task, the number of rows is capped at 2000 and the number of features is capped at 100. To reduce storage overhead, the offline pool is consumed in a round-robin manner during training (i.e., steps are iterated cyclically rather than regenerated on-the-fly).

**Unified targets for classification and regression.** When sampling the target variable from the corresponding SCM node (as proposed by Qu et al.), we additionally store two post-processed target versions for each task: (i) a *classification* target obtained by discretization, and (ii) a *regression* target obtained by normalization/standardization. This allows a single synthetic task instance to supervise both heads.

**Pretraining objective.** In each optimization step, we jointly optimize the classification and regression heads on the same batch of tasks. For classification, we use cross-entropy loss, and for regression we use a combination of mean squared error and mean absolute error.

**Pretraining schedule and compute.** We pretrain TAB-SWIFT (a 24-layer transformer with $d_{\text{model}} = 192$) for 150,000 steps using AdamW (Loshchilov & Hutter, 2019) on 8 NVIDIA RTX 5090 GPUs, which takes approximately 7 days. Following TabICL (Qu et al., 2025), after the main pretraining stage we further continue training for an additional 2,000 steps on larger tasks where the number of rows is increased up to 20,000 to improve robustness to longer in-context sequences.

**Post-training for early exit.** Starting from the pretrained

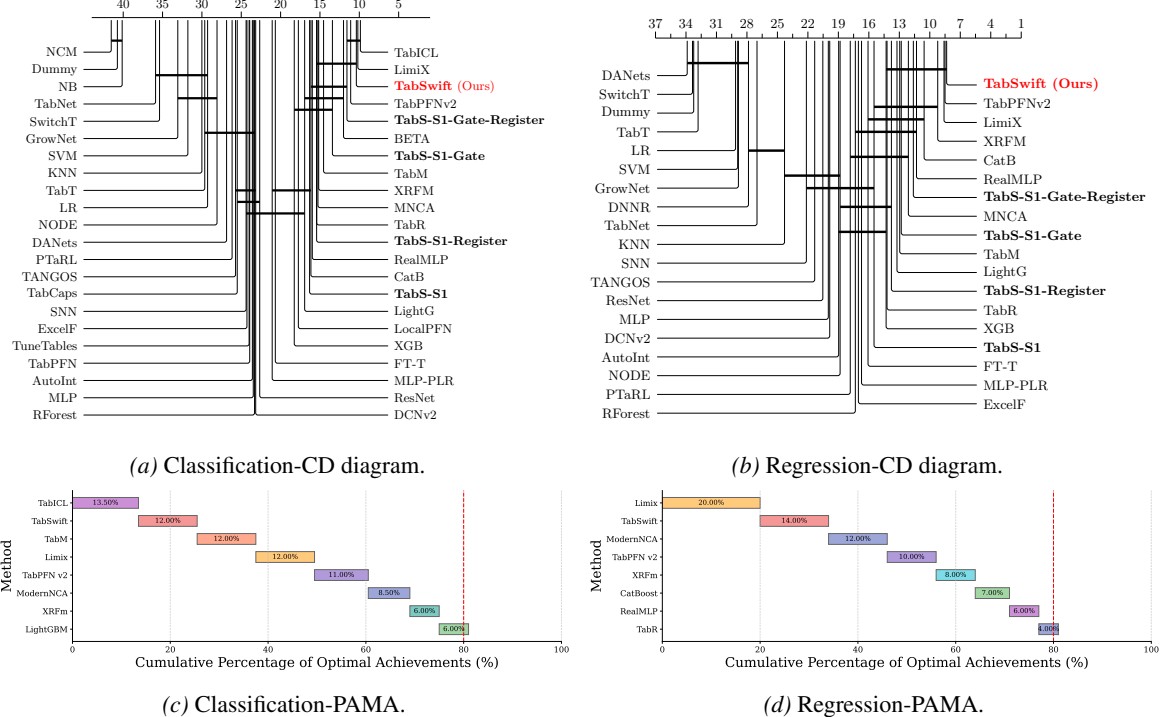

*(a)* Classification-CD diagram.

*(b)* Regression-CD diagram.

*(c)* Classification-PAMA.

*(d)* Regression-PAMA.

*Figure 4.* **Main results on classification and regression benchmarks. Top:** Critical difference (CD) diagrams summarize average ranks across datasets; methods connected by a bar are not significantly different under Wilcoxon–Holm correction ($\alpha = 0.05$). **Bottom:** PAMA reports the cumulative percentage of datasets where a method achieves the best performance.

backbone, we perform a lightweight post-training stage to enable adaptive early exiting. We attach randomly initialized layer-wise prediction heads, along with layer-wise exit heads for early-exit decisions at the designated exit layers (separate heads for classification and regression). During this stage, we freeze the pretrained backbone and train only the newly added heads for 10,000 steps, which takes about 6 hours on the same hardware. (The training objectives for the exit heads and the inference-time stopping policy are described in Section 3.2.)

### 4.3. Main Results

**Significance testing.** We compare TABSWIFT against all baselines on each dataset and perform pairwise statistical testing using the Wilcoxon signed-rank test with Holm correction (Wilcoxon–Holm) at significance level $\alpha = 0.05$. We summarize the results with critical difference (CD) diagrams, where methods are ordered by average rank (lower is better) and groups connected by a bar indicate no statistically significant difference. Overall, the CD diagrams show that TABSWIFT achieves an average rank comparable to—and in several settings better than—the strongest baselines, and it is often not significantly different from the top-performing methods under Wilcoxon–Holm correction.

**PAMA analysis.** To characterize robustness, we addition-

ally report *PAMA*, defined as the cumulative percentage of datasets on which a method attains the best performance (including ties). We compute PAMA separately for classification and regression; a larger area / faster rise indicates that a method wins on a larger fraction of datasets. Across both task types, TABSWIFT achieves a high best-achievement rate, suggesting that its strong average rank is supported by consistent performance across a substantial fraction of datasets rather than being driven by a small subset. Moreover, the PAMA profiles indicate that different tabular foundation models tend to excel on different subsets of datasets, implying complementarity; this makes TABSWIFT a promising component for model selection or ensembling with other models to further improve overall performance (Erickson et al., 2025).

**Efficiency comparison.** We measure inference time per dataset under a unified evaluation pipeline and report how runtime scales with dataset complexity. Specifically, for each dataset we record the average wall-clock time required to produce predictions for its query/test instances given the support set. For a fair comparison of backbone efficiency, we report *full-depth* inference time without early-exit (i.e., all layers are executed for every query). We sort datasets by complexity measured as $N \times d$ (number of rows times number of features) and visualize the per-dataset runtimes

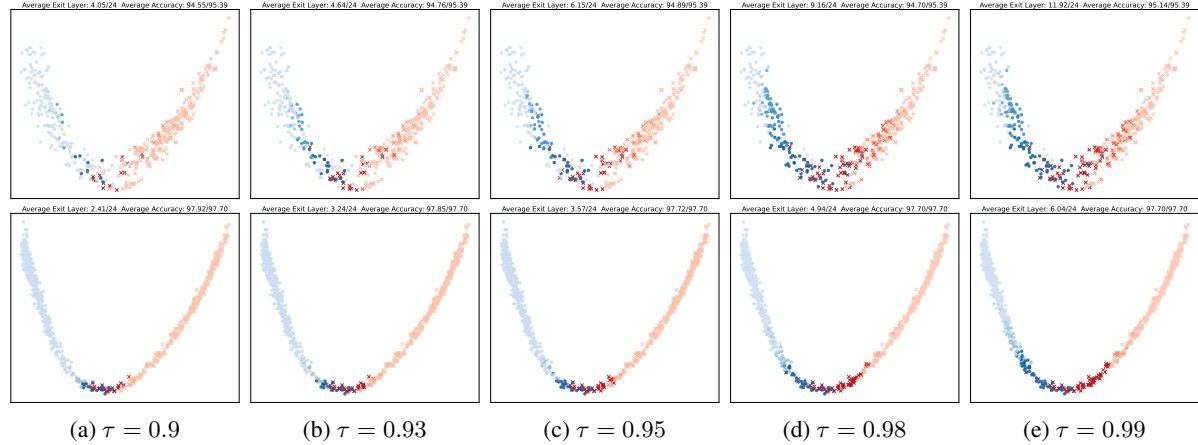

*Figure 5.* We visualize two real world binary-classification datasets: `spambase` (first row) and `twonorm` (second row) by projecting the final-layer query embeddings from TABSWIFT to 2D using PCA. Each point is a test sample; marker shape/color indicates the ground-truth class. Columns sweep the early-exit threshold $\tau$. Point opacity encodes the exit depth $\ell_{\text{exit}}$ (darker means later exit).

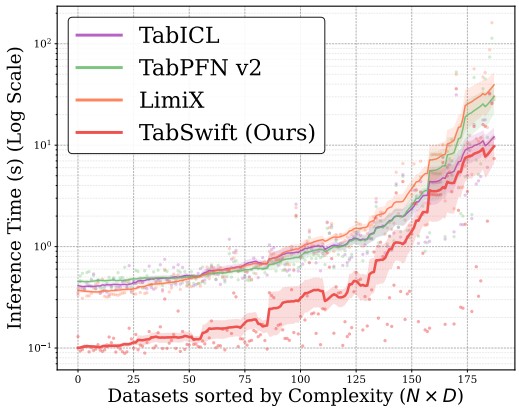

*Figure 6.* Inference time across datasets (log scale), with datasets ordered by complexity $N \times d$ (rows $\times$ features). Early-exit is disabled so that all models run at full depth for every query.

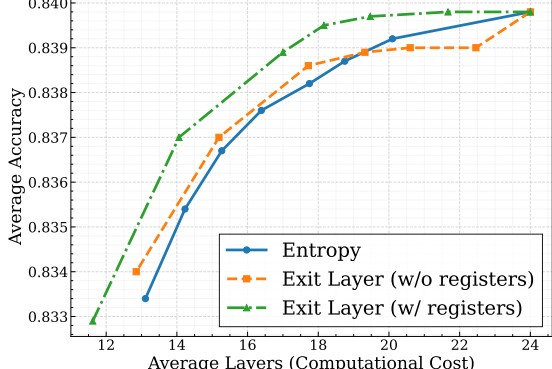

*Figure 7.* Average accuracy versus average executed layers when varying $\tau$, comparing per-sample entropy stopping and learned exit heads with/without register conditioning.

for all tabular foundation models.

## 4.4. Ablation Study

We perform ablations to quantify the contribution of each component under the same training and evaluation pipeline. To match the variants shown in Figure 4, we start from a retrained TabPFN v1-style row-wise attention-only backbone, denoted as `TabS-S1`. We then progressively add our lightweight architectural modifications: `TabS`-S1-Gate equips the backbone with element-wise gated attention, `TabS`-S1-Register prepends learnable register tokens, and `TabS`-S1-Gate-Register combines both. Finally, `TabSwift` augments `TabS`-S1-Gate-Register with two-stage pretraining, as described in subsection 4.2.

## 4.5. Adaptive Early Exit

We evaluate TABSWIFT with *per-sample* early exit by varying the exit threshold $\tau$ and report task performance (AUC/RMSE) together with the average exit depth $\mathbb{E}[\ell_{\text{exit}}]$ as a proxy for computation cost. We plot the resulting accuracy–compute Pareto curves in Figure 7. Besides our learned exit head, we include an entropy-based stopping baseline computed *per query* from the intermediate predictive distribution; this differs from (Küken et al., 2025), which uses the *test-set average* entropy at each layer to decide the exit depth (akin to a transductive / test-time-adaptation setting). We also ablate whether the exit head conditions on the pooled register representation, yielding *Exit Layer (w/o registers)* and *Exit Layer (w/ registers)*; overall, the learned exit head provides a more favorable trade-off than entropy-based stopping, and register conditioning further improves the Pareto frontier.

**Embedding-space view of per-sample early exit.** To better understand when TABSWIFT chooses to exit early, we visualize sample-wise exit depths in the representation space. For each dataset, we extract the query embeddings from the *final* layer and project them to 2D with PCA (Figure 5). We then annotate each point by the exit depth $\ell_{\text{exit}}$ produced under different thresholds $\tau$: lighter points exit earlier (shallower layers), whereas darker points require deeper computation. As $\tau$ increases, more samples defer their exit to later layers, and the deepest exits concentrate around regions where the two classes overlap in the embedding space, suggesting that the learned gate allocates extra computation to harder or more ambiguous instances while allowing clearly separable samples to terminate early. Empirically, a large fraction of samples can be reliably predicted at shallow depths, reducing average computation with negligible performance degradation.

## 5. Conclusion

We revisited the lightweight row-wise-attention-only PFN backbone and showed that, with two simple stabilizing enhancements—gated attention and a small set of learnable register tokens—it can achieve competitive performance against stronger but heavier tabular foundation models on both classification and regression. We further introduced a per-query adaptive early-exit mechanism that dynamically reduces average inference cost by exiting shallow for easy samples while preserving accuracy for hard ones. Overall, TABSWIFT offers a strong accuracy–efficiency trade-off and enables practical tabular in-context learning under latency-sensitive deployment constraints.

## Acknowledgements

This work is partially supported by Natural Science Foundation of Jiangsu Province of China under Grant (BK20250062), NSFC (62376118), the Collaborative Innovation Center of Novel Software Technology and Industrialization, the Fundamental and Interdisciplinary Disciplines Breakthrough Plan of the Ministry of Education of China (No. JYB2025XDXM118), the "111 Center" (No. B26023). We thank the authors of TabICL for releasing their training framework and data generation pipeline, which our implementation follows and builds upon.

## Impact Statement

This paper presents work whose goal is to advance the field of Machine Learning. There are many potential societal consequences of our work, none which we feel must be specifically highlighted here.

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

The Appendix consists of five sections:

# A. Related Work

**Tabular data learning.** Learning transferable predictors for tabular data has long been challenging due to heterogeneous schemas (van Breugel & van der Schaar, 2024; Yuan et al., 2025), mixed feature types, and the prevalence of small datasets. Consequently, tree-based ensembles such as gradient-boosted decision trees (GBDTs) remain strong and widely used baselines for tabular prediction (Chen & Guestrin, 2016; Ke et al., 2017; Prokhorenkova et al., 2018). A large scale of work has explored deep learning for tables via specialized architectures (e.g., attention over features (Gorishniy et al., 2021), embeddings for numerical values (Gorishniy et al., 2022), and hybrid designs (Gorishniy et al., 2025)) and extensive benchmarks and surveys have analyzed their strengths and limitations (Borisov et al., 2024; McElfresh et al., 2023; Ye et al., 2024; Erickson et al., 2025; Cheng et al., 2025).

**Tabular foundation models.** More recently, *foundation-model-style* approaches have emerged for tabular learning, aiming to amortize model selection and hyperparameter tuning across datasets. In particular, the Prior-Fitted Network (PFN) paradigm pretrains a transformer over a distribution of synthetic tasks and performs prediction through *in-context learning* (ICL) (Brown et al., 2020), directly mapping labeled examples and a query instance to a prediction without test-time parameter updates (e.g. TabPFN (Hollmann et al., 2023) and TabDPT (Ma et al., 2025)). Building on TabPFN, subsequent models expanded task coverage and improved accuracy, e.g., extending PFNs (e.g., TabPFN v2 (Hollmann et al., 2025) and LimiX (Zhang et al., 2025)) to broader settings and regression, and incorporating more sophisticated architectural components to better capture feature-wise structure and dataset heterogeneity. Parallel to this line, TabICL (Qu et al., 2025) introduces column-aware representations and multi-stage pretraining pipelines to improve scalability on larger support sets. Despite delivering substantial performance gains over TabPFN v1, these advances often come with noticeably higher inference overhead. This motivates our revisiting of a lightweight row-wise attention-only PFN backbone, improving both accuracy and efficiency through stabilization and adaptive computation rather than further increasing architectural complexity.

**Early exit and adaptive computation.** Adaptive computation aims to reduce average inference cost by allocating less computation to easy inputs and more to hard ones. A comprehensive survey on *dynamic neural networks* categorizes dynamic inference as sample-wise, spatial-wise, and temporal-wise adaptation, and highlights *dynamic depth* as a primary sample-wise mechanism (Han et al., 2022). Within dynamic depth, *early exiting* enables an input-dependent stopping layer by attaching intermediate classifiers (or cascading multiple models), so that "easy" samples can be predicted at shallow exits without executing deeper layers (Teerapittayanon et al., 2017). Existing early-exit systems typically adopt either confidence-based stopping rules (e.g., entropy/margin thresholds) or learned decision functions to determine whether to exit (Xin et al., 2020; Zhou et al., 2020; Liu et al., 2020). Related approaches also include layer skipping and other conditional execution strategies that adapt computation along depth (Graves, 2016; Fan et al., 2020). In the context of tabular foundation models, early stopping has also been explored via an entropy-based criterion that averages prediction entropies over the entire test set to decide the stopping layer (Küken et al., 2025). While effective in offline evaluation, such dataset-level stopping relies on global test-set statistics and is less suitable for realistic online deployment where predictions are made *per query*. Our method follows the early-exit paradigm but tailors the exit policy to tabular in-context learning: the gate is trained to predict *layer-wise, per-sample* prediction reliability, enabling anytime inference with accuracy–cost trade-offs.

# B. Implementation Details

**Code and checkpoints.** We will release our code and the pretrained model checkpoint publicly upon acceptance.

## B.1. Dataset Pre-processing

Unless stated otherwise, we follow the preprocessing protocol of Gorishniy et al. (2021). We standardize numerical features by subtracting the training-set mean and dividing by the training-set standard deviation. Categorical features are transformed via ordinal encoding to obtain model-compatible inputs.

## B.2. Implementations

**Methods Compared.** We compare TABSWIFT against three categories of methods to evaluate its effectiveness:

(1) **Classical Machine Learning Algorithms:** This category includes widely used classical approaches such as Support Vector Machines (SVM), K-Nearest Neighbors (KNN), xRFM (Beaglehole et al., 2025), and tree-based methods like Random Forest (RForest) (Breiman, 2001), XGBoost (XGB) (Chen & Guestrin, 2016), CatBoost (CatB) (Prokhorenkova et al., 2018), and LightGBM (LightG) (Ke et al., 2017).

(2) **Tabular Deep Learning Models:** We consider state-of-the-art deep learning models for tabular data, including TabR (Gorishniy et al., 2024), ModernNCA (MNCA) (Ye et al., 2025b), RealMLP (Holzmüller et al., 2024), TabM (Gorishniy et al., 2025), FT-Transformer (FT-T) (Gorishniy et al., 2021), MLP-PLR (Gorishniy et al., 2022), NODE (Popov et al., 2020), SNN (Klambauer et al., 2017), ExcelFormer (ExcelF) (Chen et al., 2024), BETA (Liu & Ye, 2025), LocalPFN (Thomas et al., 2024), DANets (Chen et al., 2022), and TabNet (Arik & Pfister, 2021).

(3) **Tabular Foundation Models:** Lastly, we compare with the recent TFMs, including TabPFN (Hollmann et al., 2023), TabPFN v2 (Hollmann et al., 2025), TabICL (Qu et al., 2025), and LimiX (Zhang et al., 2025).

**Ensembling protocol.** Following the analysis in (Ye et al., 2025a), increasing the number of ensembles for TabPFN v2 yields only marginal performance gains while substantially increasing inference cost. To ensure a fair efficiency comparison across methods, we therefore cap the ensemble size for TabPFN v2 and its architecturally similar variant LimiX at 4. For TabICL and TABSWIFT, we use an ensemble size of 16. Notably, even with $4\times$ more ensembles than TabPFN v2, TABSWIFT remains significantly faster in our runtime evaluation (Sec. 4.3).

**Checkpoints.** We use the official pretrained checkpoints for all foundation models. TabPFN v2 uses separate default checkpoints for classification and regression: https://huggingface.co/Prior-Labs/TabPFN-v2-clf/blob/main/tabpfn-v2-classifier.ckpt and https://huggingface.co/Prior-Labs/TabPFN-v2-reg/blob/main/tabpfn-v2-regressor.ckpt. For TabICL, we use https://huggingface.co/jingang/TabICL-clf/blob/main/tabicl-classifier-v1.1-0506.ckpt. For LimiX, we use https://huggingface.co/stable-ai/LimiX-16M/blob/main/LimiX-16M.ckpt.

**Other Methods from TALENT.** For the remaining TALENT methods, we either adopt the results reported in (Ye et al., 2024) or rerun them using TALENT's official hyperparameter configurations. When tuning is required, we perform 100 hyperparameter trials per method using the search spaces provided by TALENT[2]. Each trial is evaluated with 15 random seeds, and we report the mean performance across seeds.

**Evaluation suites.** For the main classification benchmark, we follow TALENT and evaluate on its 200 classification datasets, excluding 12 datasets with more than 10 classes. For regression, we evaluate on all 100 regression datasets in TALENT. For the adaptive early-exit experiments, we report results on the remaining 188 classification datasets (i.e., after excluding the $> 10$-class datasets) to ensure consistent stopping behavior and comparable accuracy metrics.

**Compute and hardware.** We pretrain TABSWIFT using 8 NVIDIA RTX 5090 GPUs. For evaluation, since other tabular foundation models incur substantially higher memory footprints, we run all inference-time benchmarking (including runtime measurements) on a single NVIDIA A800 GPU under the same pipeline and batch settings whenever applicable. All reported inference times are measured on this A800 setup to ensure a consistent and fair efficiency comparison across methods.

**Training Exit Heads (Auxiliary Reliability Heads).** We train a lightweight *exit head* at each exit layer to predict whether the corresponding intermediate prediction is reliable enough to stop. Concretely, for each exit layer $e$, the model outputs (i) a task prediction $\hat{\mathbf{y}}^{(e)}$ and (ii) an auxiliary logit $\hat{s}^{(e)}$ (one per decoded test position), where $\sigma(\hat{s}^{(e)}) \in (0, 1)$ is interpreted as a stopping score.

---

[2]https://github.com/LAMDA-Tabular/TALENT/tree/main/TALENT/configs

For classification, we supervise the exit head using the ground-truth correctness of the current exit's prediction. Let $\hat{y}^{(e)} = \arg\max \text{logits}^{(e)}$ be the predicted class at exit $e$, and $y$ the true label. We define the binary target

$$t_{\text{cls}}^{(e)} = \mathbb{1}\{\hat{y}^{(e)} = y\}, \tag{17}$$

and train the exit head with a binary cross-entropy loss:

$$\mathcal{L}_{\text{aux-cls}}^{(e)} = \text{BCEWithLogits}\left(\hat{s}_{\text{cls}}^{(e)}, t_{\text{cls}}^{(e)}\right). \tag{18}$$

This encourages high stopping scores exactly when the intermediate classifier is already correct.

For regression, there is no discrete notion of correctness, so we supervise exit heads by comparing the intermediate error to the final-exit error. Let $\hat{y}^{(e)}$ be the scalar prediction at exit $e$, $\hat{y}^{(E)}$ the final-exit prediction, and $y$ the ground truth. We compute per-sample absolute errors

$$\varepsilon^{(e)} = \left|\hat{y}^{(e)} - y\right|, \qquad \varepsilon^{(E)} = \left|\hat{y}^{(E)} - y\right|. \tag{19}$$

We then define a binary target indicating whether exit $e$ is not worse than the final exit up to a margin:

$$t_{\text{reg}}^{(e)} = \mathbb{1}\left\{\varepsilon^{(e)} \leq \varepsilon^{(E)} + m\right\}, \tag{20}$$

where $m > 0$ is a tunable margin (we use $m = 0.05$ under our target normalization). The corresponding auxiliary loss is

$$\mathcal{L}_{\text{aux-reg}}^{(e)} = \text{BCEWithLogits}\left(\hat{s}_{\text{reg}}^{(e)}, t_{\text{reg}}^{(e)}\right). \tag{21}$$

We apply this auxiliary objective to all exits except the final exit; the final exit has no meaningful "better-than-final" target and its auxiliary loss is omitted. Optionally, if positive targets are rare, we use a positive-class weight in BCEWithLogits to mitigate imbalance.

We train prediction heads at every exit using standard supervised losses and average them across exits. For classification,

$$\mathcal{L}_{\text{cls}} = \frac{1}{E} \sum_{e=1}^{E} \left(\mathcal{L}_{\text{pred-cls}}^{(e)} + \lambda_{\text{aux-cls}} \, \mathcal{L}_{\text{aux-cls}}^{(e)}\right), \tag{22}$$

where $\mathcal{L}_{\text{pred-cls}}^{(e)}$ is cross-entropy. For regression,

$$\mathcal{L}_{\text{reg}} = \frac{1}{E} \sum_{e=1}^{E} \left(\mathcal{L}_{\text{pred-reg}}^{(e)} + \lambda_{\text{aux-reg}} \, \mathcal{L}_{\text{aux-reg}}^{(e)}\right), \tag{23}$$

where $\mathcal{L}_{\text{pred-reg}}^{(e)}$ is a combined MSE + L1 loss:

$$\mathcal{L}_{\text{pred-reg}}^{(e)} = \left\|\hat{y}^{(e)} - y\right\|_2^2 + \left\|\hat{y}^{(e)} - y\right\|_1. \tag{24}$$

In our implementation we use weights $w_{\text{cls}} = 1.0$, $w_{\text{reg}} = 0.5$, $\lambda_{\text{aux-cls}} = 0.5$, and $\lambda_{\text{aux-reg}} = 1.0$.

## C. Limitations

Our study focuses on supervised tabular prediction, primarily classification and regression, and does not evaluate other structured-data settings such as ranking, survival analysis, multi-label prediction, or time-series forecasting. In addition, TABSWIFT is pretrained on synthetic task distributions; while this enables broad coverage and strong out-of-the-box performance, the match between synthetic priors and a target domain may affect performance on highly specialized datasets.

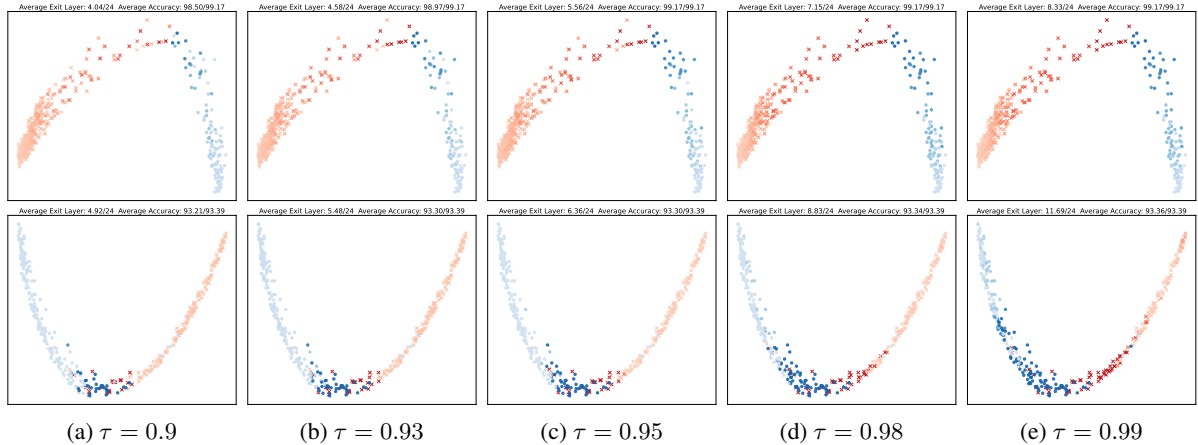

(a) $\tau = 0.9$   (b) $\tau = 0.93$   (c) $\tau = 0.95$   (d) $\tau = 0.98$   (e) $\tau = 0.99$

*Figure 8.* **Early-exit behavior visualized in the embedding space (additional datasets).** We visualize two real-world binary-classification datasets, Long (first row) and rice_cammeo_and_osmancik (second row), by projecting TABSWIFT's *final-layer* representations of the query/test tokens to 2D with PCA. Each point corresponds to a test sample; marker shape/color indicates the ground-truth class. Columns sweep the early-exit threshold $\tau$ used by the per-sample exit head. Point opacity encodes the exit depth $\ell_{\text{exit}}$ (lighter $\rightarrow$ earlier exit; darker $\rightarrow$ later exit, i.e., more layers executed). As $\tau$ increases, the exit policy becomes more conservative and more samples defer to deeper layers, with later exits frequently occurring for samples near class overlap regions in the embedding space.

*Table 1.* **Per-sample entropy baseline: accuracy–compute trade-off under different thresholds.** We report the average accuracy and the average executed layers when using a per-query entropy stopping rule. Full denotes running all $L=24$ layers without early exit. Smaller thresholds are more conservative (deeper execution) and typically yield slightly higher accuracy. These numbers correspond to the *Entropy* curve in Figure 7.

| Threshold | full | 0.01 | 0.05 | 0.1 | 0.2 | 0.3 | 0.5 |
|---|---|---|---|---|---|---|---|
| Avg. Accuracy | 0.8398 | 0.8392 | 0.8387 | 0.8382 | 0.8376 | 0.8367 | 0.8334 |
| Avg. Layers | 24 | 20.1 | 18.75 | 17.75 | 16.39 | 15.27 | 13.11 |

# D. Additional Results

To provide additional intuition for the learned per-sample exit policy, we visualize TABSWIFT's final-layer query/test embeddings on two extra binary-classification datasets and overlay each sample's exit depth under different thresholds $\tau$ (Figure 8).

To complement Figure 7, we report the underlying threshold sweeps in Tables 1–3. Each table corresponds to one curve in Figure 7: Table 1 summarizes the per-sample *entropy* baseline (using a per-query confidence threshold), Table 2 reports the learned *exit head* without register conditioning, and Table 3 reports the learned *exit head* with register conditioning. Across all three settings, tightening the threshold leads to deeper computation and slightly higher accuracy, revealing a clear accuracy–compute trade-off.

We also apply our per-sample early-exit mechanism to regression and report the corresponding trade-off on the 100 TALENT regression datasets in Figure 9. Consistent with the classification results, tightening the stopping criterion (larger $\tau$) shifts more samples to deeper exits, increasing the average executed layers and improving regression quality (higher $R^2$). This suggests that the learned exit head provides a controllable accuracy–compute knob beyond classification, enabling adaptive inference for continuous prediction tasks as well.

**Performance Beyond 20,000 Training Instances.** We further examine the behavior of TABSWIFT when the number of training instances exceeds the range considered during continued pretraining. This is an important setting because TABSWIFT, as a PFN-style in-context learner, relies on row-wise self-attention, whose computational cost grows quadratically with the number of rows. Therefore, very large-$N$ regimes can eventually become challenging compared with methods explicitly designed for large-sample tabular learning.

Our current study primarily targets the small-to-medium tabular prediction regime represented by PFN-style in-context learning, and the continued pretraining of TABSWIFT is conducted on datasets with up to 20k rows. We additionally evaluate

*Table 2.* **Learned exit head (w/o registers): accuracy–compute trade-off under different $\tau$.** Average accuracy and average executed layers when applying the learned per-sample exit head without register conditioning. `Full` denotes full-depth inference ($L{=}24$). Higher $\tau$ enforces stricter exiting, increasing the average depth and improving accuracy marginally. These numbers correspond to the *Exit Layer (w/o registers)* curve in Figure 7.

| Threshold | full | 0.999 | 0.995 | 0.99 | 0.98 | 0.95 | 0.9 |
|---|---|---|---|---|---|---|---|
| Avg. Accuracy | 0.8398 | 0.8390 | 0.8390 | 0.8389 | 0.8386 | 0.8370 | 0.8340 |
| Avg. Layers | 24 | 22.46 | 20.6 | 19.32 | 17.72 | 15.19 | 12.85 |

*Table 3.* **Learned exit head (w/ registers): accuracy–compute trade-off under different $\tau$.** Average accuracy and average executed layers when using the learned per-sample exit head conditioned on the pooled register representation. `Full` denotes full-depth inference ($L{=}24$). As $\tau$ increases, the model exits later on average and attains higher accuracy. These numbers correspond to the *Exit Layer (w/ registers)* curve in Figure 7.

| Threshold | full | 0.999 | 0.995 | 0.99 | 0.98 | 0.95 | 0.9 |
|---|---|---|---|---|---|---|---|
| Avg. Accuracy | 0.8398 | 0.8398 | 0.8397 | 0.8395 | 0.8389 | 0.8370 | 0.8329 |
| Avg. Layers | 24 | 21.67 | 19.47 | 18.16 | 17.0 | 14.06 | 11.62 |

TABSWIFT on TALENT classification datasets with more than 20,000 training samples. Table 4 reports the average rank of different methods in this large-$N$ subset.

*Table 4.* Average rank on TALENT classification datasets with more than 20,000 training samples. Lower is better.

| Method | RealMLP | ModernNCA | TabM | TABSWIFT | TabICL | CatBoost | LightGBM | TabPFN v2 |
|---|---|---|---|---|---|---|---|---|
| Avg. Rank | 9.31 | 9.69 | 11.65 | 12.32 | 12.56 | 16.04 | 16.10 | 16.61 |

The results show that TABSWIFT remains competitive with PFN-style and transformer-based baselines such as TabICL and TabPFN v2, but its relative advantage becomes less pronounced once the number of training instances moves beyond the row range used in continued pretraining. In this regime, non-PFN tabular methods such as RealMLP, ModernNCA, and TabM can become more favorable. We therefore regard scaling PFN-style in-context learners to larger-$N$ datasets as an important limitation of the current work and a promising direction for future research.

**Performance on High-Dimensional Datasets.** We further evaluate TABSWIFT on high-dimensional tabular datasets, where the feature dimension exceeds the fixed input budget of the PFN-style backbone. This is an important practical concern because TABSWIFT uses a fixed-dimensional input representation. In our implementation, PCA is applied only when the original feature dimension is larger than the input budget, and the feature dimension is reduced to 100 in such cases.

We additionally evaluate on 9 high-dimensional datasets considered in Ye et al. and Liu & Ye. To ensure reliable evaluation, we select datasets with more than 150 samples. Following the evaluation protocol of the original paper, we use 5 train-test splits and 3 random seeds per split. We compare TABPFN V2, TABSWIFT, TabICL, and XGBoost. To remain consistent with our main benchmark protocol, TABSWIFT and TabICL use 16-model ensembles, while TABPFN V2 uses a 4-model ensemble, as in our main experiments. The detailed results are reported in Table 5.

The results show that TABSWIFT remains competitive on high-dimensional datasets, achieving the second-best average rank among the compared methods. Although its performance is overall somewhat weaker than on standard tabular benchmarks, we do not observe a clear monotonic degradation trend as the feature dimensionality increases.

These results suggest that high-dimensional datasets remain challenging for current TFM-based methods, but PCA-based dimensionality reduction provides a practical and effective way to adapt fixed-input PFN-style models to this regime. Nevertheless, we acknowledge that very high-dimensional problems are not the primary target setting of TABSWIFT, and improving feature encoding or dimensionality reduction for such datasets is an important direction for future work.

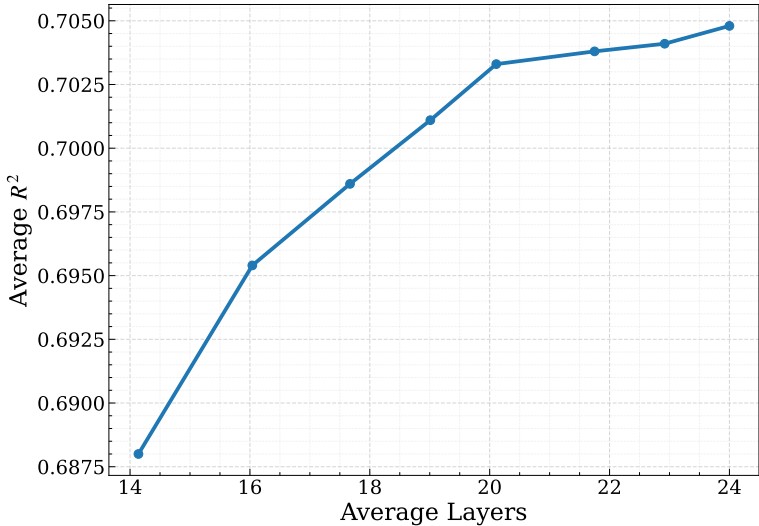

*Figure 9.* **Regression early-exit trade-off.** Average $R^2$ versus the average executed layers on the 100 TALENT regression datasets when sweeping the per-sample early-exit threshold $\tau$. Larger average depth corresponds to stricter stopping (larger $\tau$), yielding higher accuracy at increased compute.

*Table 5.* Performance on high-dimensional datasets. The best result on each dataset is highlighted in bold.

| Dataset | #Dim. | TABPFN V2 | TABSWIFT | TabICL | XGBoost |
|---|---|---|---|---|---|
| warpPIE10P | 2420 | **1.00** | **1.00** | **1.00** | 0.95 |
| PCMAC | 3289 | **0.93** | 0.92 | 0.89 | 0.92 |
| lung | 3312 | **0.95** | 0.92 | **0.95** | 0.94 |
| RELATHE | 4322 | 0.86 | **0.91** | 0.87 | 0.87 |
| BASEHOCK | 4862 | **0.98** | **0.98** | 0.95 | 0.95 |
| gisette | 5000 | 0.97 | **0.99** | 0.96 | 0.98 |
| TOX_171 | 5748 | 0.81 | 0.79 | **0.83** | 0.78 |
| arcene | 10000 | **0.84** | 0.81 | 0.79 | 0.75 |
| SMK_CAN_187 | 19993 | **0.71** | 0.70 | 0.69 | 0.66 |
| Avg. Rank | – | **1.8** | 2.1 | 2.9 | 3.2 |

# E. The Use of Large Language Models

We used a large language model (LLM) solely as a writing assistant to improve the clarity and readability of the manuscript (e.g., polishing grammar and phrasing). The LLM was not involved in research ideation, experimental design, implementation, or analysis. All scientific contributions and results are entirely the work of the authors.

