# OpenReview forum: "TabSwift: An Efficient Tabular Foundation Model with Row-Wise Attention"
_ICML.cc/2026/Conference — ICML 2026 spotlight_

### Official Review · Reviewer_5kin · 2026-03-06

**Soundness:** 2
**Presentation:** 3
**Significance:** 2
**Originality:** 2
**Overall Recommendation:** 4
**Confidence:** 3

**Summary:**

This work improves the existing work TabPFN by integrating a gated attention stabilization mechanism and a small set of learnable register tokens. The proposed methods show the best performance and the lowest inference time.

**Compliance With Llm Reviewing Policy:**

Affirmed.

**Final Justification:**

The rebuttal fully addresses my concern. Hence, I increased the score.

**Key Questions For Authors:**

1. When increasing $\tau$, the exit depth increases. The performance is presumably better. However, in Figure 5 second row, the average accuracy decreases with increasing $\tau$. Can the authors explain this phenomenon?

**Limitations:**

yes (In Appendix)

**Strengths And Weaknesses:**

## Strengths

1. The proposed method shows good performance in terms of both efficiency and effectiveness.

2. The time complexity analysis is helpful for explaining the higher efficiency of SwiftPFN (using row attention only) compared to models applying alternating row/column attention.

3. Proposed designs and their differences compared to existing related works are clearly described.

## Weaknesses

1. The difference between the proposed learnable register tokens and the existing work is unclear

Although the related work (i.e. TabPFN-2.5) to the idea of learnable register tokens is mentioned, the difference between the existing work and this work is unclear. Since the idea of learnable register tokens is claimed as one of two main contributions of this work, it is of great importance to elaborate on the differences.

2. TabPFN-2.5 has a similar design of introducing latent tokens, which is also mentioned in the manuscript. However, in the performance comparison (Figure 4), this baseline is deliberately missing. Please include this method as a baseline.

3. There are some issues in Figure 4 (main results of comparing performance with baselines).

3.1. Performance in Figure 1 seems to be contradictory to the performance in Figure 4. In Figure 4, SwiftPFN does not show a significantly better performance for both classification and regression tasks. On one side, both TabICL and Limix show better performance than SwiftPFN on the classification tasks. On the other side, the Wilcoxon signed-rank test indicates that there are many baselines that have statistically similar performance compared to SwiftPFN. Figure 1 shows that when combining both classification tasks and regression tasks, SwiftPFN exhibits much better performance than baselines as the average rank of SwiftPFN is close to 1 while the best baselines have an average rank close to 6. Can the authors explain this discrepancy?

3.2. The performance of TabICL is shown in the classification task (Figure 4 (a) and (c)) yet is missing in the regression task (Figure 4 (b) and (c)).

3.3. Despite missing a strong bassline TabICL in the regression task, the average performance on the PAMA analysis indicates that Limix shows better average performance than SwiftPFN (16% vs 13%).

4. Some issues/questions in Figure 5.

4.1. The average accuracy is reported in the format of $p_1/p_2$. However, there is no place to explain what it means.

4.2. In the current sweep range for $\tau$ is [0.9, 0.99], the average accuracy is similar. When further decreasing $\tau$, will there be a more pronounced performance drop?

5. An important experimental detail is missing.

5.1. Introducing a set of learnable register tokens is one of two key contributions of this work. However, there is no place to report the number of learnable register tokens, i.e., what is the value of $K$ used in this work.

5.2. Besides, there is no ablation study to show the effect of different $K$ values.

6. Figure 1 is unclear.

Without a text description, I guess, by legend, that bubble 2 corresponds to the performance when a small set of learnable register tokens is added and bubble 3 means the performance when the gated attention stabilization mechanism is applied. How is the performance of the SwiftPFN obtained? SwiftPFN improves TabPFN by those two components, which seem to correspond to bubble 3.




[1] Küken, Jaris, Lennart Purucker, and Frank Hutter. "Early stopping tabular in-context learning." arXiv preprint arXiv:2506.21387 (2025).

---

> ### Author Rebuttal · Authors · 2026-03-31
>
> Thank you for the careful reading and constructive feedback. We address the main concerns below.
>
> > Q1 Register tokens and relation to prior work
>
> We agree that the connection to prior work, especially TabPFN v2.5, should be explained more clearly. Our intention is not to claim register tokens as a standalone invention. In our work, their main role is to **aggregate global information from the entire dataset**, which is particularly helpful for strengthening a lightweight row-attention-only PFN backbone. We also note that TabPFN v2.5 includes related tokens, but the discussion there is mainly framed around **additional computation** and **LLM attention sinks**. We will clarify this more explicitly in the final version.
>
> > Q2 Additional stronger baselines
>
> We agree that stronger baselines should be discussed more explicitly. We additionally included TabDPT v1.1; in the Fig. 4-style comparison, its average rank is around 13.4.
>
> Regarding TabPFN v2.5, we agree that it is a strong recent baseline. At the same time, although the checkpoint is available, some details of the full pretraining pipeline are not available at the same level of control as for fully open training/data setups. Therefore, in our main benchmark, we focus more on models whose training/data setup is more reproducible, such as TabICL. More broadly, our goal is not simply to maximize raw performance, but to study how far a lightweight row-attention-only PFN backbone can be pushed while keeping inference cost low. As noted in Sec. 4.2, our pretraining still uses the earlier data engine; despite this, SwiftPFN already achieves competitive performance with strong open baselines using a substantially lighter architecture. We believe this suggests meaningful headroom for the row-attention-only route, and our modifications are largely orthogonal to newer systems. We will discuss this more explicitly in the final version.
>
> > Q3 Clarifying Figs. 1 and 4
>
> We agree that the current presentation can make Figs. 1 and 4 look more contradictory than they actually are. Fig. 1 is a compact combined summary / trade-off view, while Fig. 4 reports task-specific rank-based comparisons and significance tests for classification and regression separately.
>
> For visual clarity, Fig. 1 only includes a subset of methods, rather than the full method list. Methods such as KNN, SVM, ExcelFormer, and AutoInt were omitted from Fig. 1, since otherwise the gap between TabPFN v1 (with the checkpoint of the original TabPFN paper, ICLR 2023) and the other TFM methods would become too large and make the figure much harder to read. We agree this should have been stated explicitly, and we will either explain the subset in the appendix or redraw Fig. 1 more clearly in the final version.
>
> The regression result is absent because TabICL v1.1 does not support regression tasks. Regarding PAMA, we agree that LimiX shows a higher average improvement than SwiftPFN on that specific analysis. This is consistent with our broader claim that SwiftPFN is not uniformly best on every single analysis, but achieves a strong overall balance of accuracy and efficiency with a lightweight row-attention-only design.
>
> > Q4 Clarifying Fig. 5
>
> Specifically, p1 denotes the dataset accuracy under the current early-exit threshold tau, while p2 denotes the dataset accuracy when running through all layers without early exit.
>
> The current tau range was chosen to cover the most practical operating region. If tau is decreased further, then almost all samples exit in very shallow layers and the performance drops to a level that is no longer practically meaningful.
>
> For the question of why the second-row average accuracy can decrease when tau increases: the row currently shows a single dataset, chosen mainly for visualization clarity. Since this dataset is relatively easy, deeper exits do not necessarily improve accuracy monotonically on that dataset. By contrast, as shown in Fig. 7, after averaging accuracy over all classification datasets, the overall trend is indeed that using more layers improves performance while also increasing inference time. We will clarify this distinction in the final version.
>
> > Q5 Missing implementation details
>
> We agree that these details should be reported explicitly.
>
> - The number of register tokens in our main experiments is 32.
> - We also agree that an ablation over the number of register tokens would be useful. Under a lightweight preliminary setup, we compared 8 / 16 / 32 / 64 register tokens by training from scratch for 10k steps on classification synthetic tasks. The classification loss decreases from 0.7852 → 0.7837 → 0.7813 → 0.7804, suggesting that more register tokens can modestly improve performance while also slowing inference. We will discuss this trade-off more explicitly in the final version.
>
> We will also clarify Fig. 1 more explicitly: (1) is SwiftPFN without gated attention and without register tokens, (2) adds register tokens, and (3) full SwiftPFN.

---

> > ### Author Rebuttal · Reviewer_5kin · 2026-03-31
> >
> > I thank the authors for addressing part of my concerns.
> >
> > 1. Only comparing SwiftPFN and LimiX methods, Figure 1 shows there is a remarkable performance difference. However, in Figure 4, SwiftPFN and LimiX have similar performance.

---

> > > ### Author Response · Authors · 2026-04-01
> > >
> > > Thank you for the follow-up. In Fig. 1, the purple bubble corresponds to LimiX, while SwiftPFN is the orange marker labeled “3”, as mentioned at the end of **Q5**. We suspect there may have been some confusion with the star-shaped marker: it does not denote our method. Instead, following the plotting convention used in the TALENT benchmark [1], the star indicates the ideal model, i.e., the hypothetical best point with both strongest predictive performance and lowest inference time. We will make this annotation clearer in the final version. The **x-axis** already reflects predictive performance via **average rank**, and under this metric **LimiX and SwiftPFN are indeed close, consistent with Fig. 4**. The more noticeable separation comes mainly from the **y-axis** (**average inference time**), where **LimiX is slower than SwiftPFN**. So Fig. 1 is not showing a large predictive-performance gap between SwiftPFN and LimiX; it is showing a similar-accuracy but clearly different *accuracy-efficiency trade-off*. We will clarify this more explicitly in the final version.  We hope this addresses your concern and merits an updated score. We sincerely appreciate the time and effort you devoted to reviewing our work.
> > >
> > > [1] A Closer Look at Deep Learning Methods on Tabular Datasets.

---

### Official Review · Reviewer_yeTt · 2026-03-09

**Soundness:** 3
**Presentation:** 3
**Significance:** 3
**Originality:** 3
**Overall Recommendation:** 5
**Confidence:** 4

**Summary:**

The paper introduces SwiftPFN, an efficient tabular foundation model that only uses a TabPFNv1-like architecture with row-based tokens instead of cell-based tokens. By adding gated attention and register tokens, it reaches an accuracy comparable to TabPFNv2 and TabICL on the TALENT benchmark while providing much faster inference. Moreover, a sample-wise early exit mechanism allows to dynamically skip some of the last layers for some samples with a small accuracy hit. Ablations show the benefit of the proposed additions.

**Compliance With Llm Reviewing Policy:**

Affirmed.

**Final Justification:**

SwiftPFN shows that with two simple architectural modifications and some smaller optimizations (24 layers, lower width, joint classification and regression training), a TabPFNv1-style architecture can achieve surprisingly good results despite being much faster than current TabPFNv2-style architectures. While this is not the most original part, it is technically solid and can be useful for further model improvements. The additional early exit mechanism adds to the novelty, even though it might be less of an improvement. The authors have addressed my concerns in the rebuttal and I think the added ablations will be interesting to the community.

**Key Questions For Authors:**

(Q1) It is interesting that the authors created a shared classification + regression model; they could mention that LimiX did the same. (An ablation on whether the joint pre-training on both helps could be interesting, as LimiX did not do one.) In l. 289 (right) it sounds like you can train both classification and regression head with the same synthetic dataset, however, you would still need to provide different versions of the target in the query sets for both tasks, right? It was also unclear to me why you say you generate 20,000 pretraining steps but then pretrain for 150,000 steps; do you use multi-epoch pre-training?

**Limitations:**

There is a short discussion of limitations in Appendix C, but it could be extended (e.g., no distributional regression, probably not great scaling to many features, what about missing value handling?, limitations of early-exit for batched inference, etc.).

**Strengths And Weaknesses:**

**Strengths**:
- Significance: Tabular foundation models are a popular and practically relevant topic, and improving their training and inference times is important for increasing their adoption.
- Presentation: The paper is well-written and includes interesting ablations.
- An open-source code release is promised.

**Weaknesses**:

(W1) Baselines: The paper mentions TabDPT as another model that only relies on row-based tokens, but does not include it in the evaluation. (For larger datasets, when using the retrieval mechanism of TabDPT, the inference time can be large due to the conflict with batching, but it would still be interesting to see a comparison.) Moreover, while TabPFN-2.5 is of course hard to compete with, its omission in the benchmark conveys the wrong impression that SwiftPFN is competitive with the state of the art.

(W2) The early exit is advertised as speeding up inference, but (1) evaluating the exit layers also comes with costs (though with lower complexity), and the practical inference time with early-exit is not evaluated; (2) in a batched inference setting without KV-caching, the self-attention on train (which often dominates runtime) needs to be run until all of the test samples have exited, which might be at the last layer; (3) it would be nice to see how big the deteriorations in e.g. Figure 7 are compared to other methods (e.g. by looking at the average rank on TALENT, or by putting some other methods as horizontal lines in the plot).


Overall, I think registers + gated attention are simple drop-ins but the insight and demonstration that they work on such a cheap architecture is valuable and the fact that they are simple makes them more attractive to build on; moreover, the architecture is further optimized with more layers and shared classification + regression. The early exit seems more original but also less clearly useful. I'm ready to update my score if some weaknesses can be addressed.

**Comments** (not necessary to address all of them in the rebuttal):
- l. 319: Liu et al 2025 is the TALENT toolbox paper, but the text refers to the benchmark (Ye et al 2024).
- The authors emphasize the lower runtime complexity of SwiftPFN, but with the PCA for large numbers of features, the complexity becomes O(N^2 + min(N^2d, Nd^2)) which for d < N is equivalent to the O(N^2 + Nd^2) of TabICL. In that sense, the advantage of SwiftPFN is more in the constant.
- I think the authors could sell their method better by also evaluating CPU inference times, and by mentioning that the low memory requirements of SwiftPFN also facilitate KV-caching for larger datasets without OOM.
- I am surprised by the mixture of MSE and MAE for regression (some details could be provided). Adding in MAE means the optimal prediction isn't the conditional mean P(y|x) anymore, which should be suboptimal when the model is evaluated with MSE on noisy data?
- I did not find how many register tokens were used.
- It could be nice to mention how many parameters SwiftPFN has.
- The authors only mention late in the paper that register tokens were introduced in TabPFN-2.5, such that in the beginning it sounds like a novel contribution.

---

> ### Author Rebuttal · Authors · 2026-03-31
>
> Thank you for the thoughtful feedback and for recognizing the value of improving efficient tabular foundation models. Below we address the main concerns:
>
> > Q1 About early exit
>
> We agree that early exit should be positioned more carefully: it is mainly designed for the online / per-query setting where test samples arrive one by one, not for all inference settings
> Following the reviewer’s suggestion, we measured per-query inference time (ms) in online setting with a fixed support size of 20000:
>
> ||w/o cache|w/ cache|
> |-|-|-|
> |no-early-exit|798|10.18|
> |Layer 18|577|8.67|
> |L 12|374|5.84|
> |L 6|171|3.30|
>
> Early exit reduces inference time in the online setting, though the gain is not perfectly proportional because exit evaluation also adds a small cost.
>
> In the batch-inference setting without KV-Cache, train-side attention may still need to be computed until the last remaining sample exits, so the realized wall-clock gain can be much smaller. We will revise the paper to make this limitation explicit. But when combined with KV caching and suitable implementation support, samples that have already exited need not participate in later-layer computation, which can still reduce inference cost in batched settings. Exploring such optimized batched early-exit implementations is an interesting direction for future work, whereas the current paper mainly focuses on the streaming / one-query-at-a-time setting.
> We will also improve this discussion and add more rank comparison in the final version.
>
> > Q2 Clarifications on implementation details
>
> Thank you for these suggestions. In the final version, we will clarify that:
>
> * We will correct the citation around the TALENT benchmark.
> * We will discuss the PCA time cost more explicitly in the final version.
> * SwiftPFN uses 32 register tokens and has about 8.2M parameters.
>
> Our motivation for register tokens is to aggregate global dataset information for a lightweight row-attention-only backbone. We also note that TabPFN v2.5 includes related tokens (“thinking rows”), but its discussion is mainly framed around additional computation and LLM attention sinks. We will strengthen this discussion in the final version.
>
> > Q3 Additional baselines
>
> Following the reviewer’s suggestion, we additionally included TabDPT v1.1, whose average rank in the Fig. 4-style comparison is about 13.4.
>
> Regarding TabPFN v2.5, we agree that it is a strong recent baseline and should be discussed more explicitly. At the same time, although its checkpoint is available, some details of the full pretraining pipeline are not available at the same level of control as for fully open training/data setups. Therefore, in our main benchmark, we focus more on models whose training/data setup is more transparent and reproducible, such as TabICL.
>
> More broadly, our goal is not simply to pursue the strongest possible architecture at any cost, but to study how far a lightweight row-attention-only PFN backbone can be pushed while keeping inference cost low. As noted in Sec. 4.2, our pretraining still uses the earlier TabICL data engine; despite this, SwiftPFN already achieves competitive performance with strong open baselines. We believe this suggests that the row-attention-only route still has meaningful headroom, and that our proposed modifications are largely orthogonal to newer models. We will discuss this more explicitly in the final version.
>
> > Q4 Regression loss formulation
>
> Thank you for asking for clarification. Please see Q6 of our response to Reviewer hqtJ.
>
> In early experiments, we tried MSE only, MAE only, and their combination, and found the mixed objective to work better. We conjecture this is partly because our current synthetic regression data are somewhat noisy, making MAE helpful for robustness while MSE still provides a useful optimization signal.
>
> > Q5 Shared classification + regression pretraining and relation to LimiX, and the 20k-vs-150k wording
>
> Thank you for highlighting this point. We agree that the relation to LimiX should be discussed more explicitly, and we will add that discussion.
>
> We also agree that the wording around “20,000” versus “150,000” is easy to misread. Due to resource constraints, we could not offline store enough synthetic datasets for all 150k optimization steps, so we generated data for 20k steps and then replayed/reused them during pretraining.
>
>
> Regarding the query targets: yes, for the same synthetic dataset, we construct both two task versions of the target.
>
> Due to time constraints, we ran a small preliminary comparison over 10k steps: classification-only, regression-only**, and joint classification + regression training. After EMA, joint training yields slightly lower losses (0.7795 vs. 0.7813 for classification; 0.7027 vs. 0.7058 for regression), suggesting that learning both views of the same data provides more complete supervision and effectively increases batch diversity/size.

---

> > ### Author Rebuttal · Reviewer_yeTt · 2026-04-02
> >
> > I thank the authors for their detailed response, which (mostly) addresses my concerns. I appreciate the new experiments on the runtime, TabDPT, and the ablation on joint training. I will raise my score to 5 (but I first need to write a final justification to update).
> >
> > The answers to Q5 are not fully clear to me. Do you generate 20k samples once and then do multiple epochs, or do you generate 20k, train on 20k, delete the 20k, generate another 20k, train 20k, ...?
> >
> > Re joint training: So you generate a dataset (X, y_class, y_reg) and then during pre-training do two forward passes, one on (X, y_class) with the classification encoder + head, and one on (X, y_reg) with the regression encoder + head? And for the comparison over 10k steps, does that mean the joint training only sees half the number of classification datasets compared to the 10k steps for the classification-only model?

---

> > > ### Author Response · Authors · 2026-04-03
> > >
> > > Thank you very much for the follow-up and for the positive update.
> > >
> > > For the first question: we first generate a pool corresponding to **20k pretraining steps** (following **TabICL**, each step corresponds to a batch of synthetic datasets), and then train for **150k steps by sampling with replacement from this fixed pool**. In other words, we do **not** keep generating a fresh new 20k-step pool after each block; instead, the same pool is replayed/reused throughout pretraining.
> > >
> > > For the second question: your understanding of the first part is exactly right. For one synthetic dataset, we construct both a **classification target** and a **regression target**, and in one training step we compute both losses and sum them. Under the **10k-step** comparison, the joint model therefore sees the **same number of classification datasets** as the classification-only model, while additionally receiving the same amount of regression supervision on the same data.
> > >
> > > So relative to classification-only training, joint training does not reduce the amount of classification data seen; it adds regression supervision on top of the same classification exposure. Its total compute is thus closer to training separate **classification** and **regression** models for the same number of steps than to halving the amount of classification training.
> > >
> > > This is also why we think the gain from joint training likely comes from more complete supervision on the same data and an effectively larger task-view batch. We will clarify these training details more explicitly in the final version. Thank you again for the careful reading and for considering an updated score. We sincerely appreciate your valuable contribution to improving the quality of our work.

---

### Official Review · Reviewer_hqtJ · 2026-03-11

**Soundness:** 3
**Presentation:** 3
**Significance:** 3
**Originality:** 2
**Overall Recommendation:** 6
**Confidence:** 4

**Summary:**

SWIFTPFN is a lightweight Tabular Foundation Model (TFM) that builds upon the Prior-Fitted Network (PFN) paradigm. The authors revisit the minimalist row-wise attention-only architecture (originally seen in TabPFN) and demonstrate that it can remain highly competitive with more complex, computationally expensive models like TabPFN v2 or TabICL.

As I understand, the model introduces three primary enhancements:

1.
**Architecture:** It stabilizes the row-wise attention backbone using **element-wise gated attention** and introduces **learnable register tokens** to provide global task-level context.


2.
**Adaptive Early Exit:** It implements a per-query, layer-wise early-exit mechanism that uses "exit heads" to dynamically determine if a prediction at a shallow layer is reliable enough to stop.


3.
**Unified Pretraining:** Unlike models that specialize in one task, SWIFTPFN uses a unified pretraining formulation that supports both **classification and regression** within a single model.



Experiments on the TALENT benchmark (300 datasets) show that SWIFTPFN achieves an average rank comparable to the strongest baselines while offering significantly better inference efficiency.

**Compliance With Llm Reviewing Policy:**

Affirmed.

**Final Justification:**

The rebuttal adresses all of my concerns and also the other reviewers concerns were adressed adequately. Hence I increased the score.

**Key Questions For Authors:**

### **4. Questions for the Authors**

1.
**The $d_n$ Term in Attention:** In Equation 1, you define the attention scaling factor as $\sqrt{d_n}$. However, standard scaled dot-product attention typically uses the square root of the head dimension ($d_h$). Is $d_n$ a typo for $d_h$, or is this a specific scaling choice intended to account for the number of rows?


2. **Performance on Large Datasets:** How does the model perform as the number of instances ($N$) grows significantly beyond the 20,000-row limit used during the continued pretraining stage? Is there a point where the quadratic complexity of row-wise attention becomes a bottleneck compared to linear attention variants?


3. **High-Feature Datasets:** Since you use PCA for datasets where the feature count $F$ exceeds $F_{max}$, have you observed a significant performance drop on "wide" datasets compared to models that use column-wise attention or more sophisticated feature tokenization?


4. **Comparison to TabICL v2:** While TabICL v2 is a very recent development, how does SWIFTPFN’s row-wise approach compare to its updated architecture in terms of handling highly heterogeneous data? (Since TabICL was released after the deadline, this does not affect the score and is purely out of interest).
5. **Regression Loss:** Could you clarify the specific weighting or formulation used for the regression loss? You mention a "combination of mean squared error and mean absolute error" - is this a simple sum, or is there a specific schedule or weighting applied to these terms during joint training with classification?

**Limitations:**

yes

**Strengths And Weaknesses:**

### **Strengths**

*
**Efficiency-Accuracy Trade-off:** By maintaining a row-wise attention structure, the model avoids the $\mathcal{O}(d \cdot n^2 + n \cdot d^2)$ cost associated with alternating row/column attention models, making it much faster for practical deployment.


*
**True Per-Query Inference:** The early-exit mechanism is designed for **online inference**, meaning it makes decisions based solely on the current query and register tokens rather than requiring a full test batch for statistics.


*
**Stabilized Training:** The addition of gated attention and register tokens addresses the capacity limitations of pure row-wise attention, allowing the model to bridge the performance gap with heavier "column-aware" architectures.


*
**Versatility:** A single backbone serving both classification and regression tasks simplifies the deployment pipeline and suggests a robust general-purpose tabular representation.



---

### **3. Weaknesses**

*
**Incremental Novelty:** The individual components (gated attention, register tokens, and early exits) are established techniques in LLMs and Vision Transformers; the novelty is primarily in their successful integration into the PFN framework.


*
**Dimensionality Handling:** For datasets with a very large number of features ($F > F_{max}$), the model relies on **PCA projection**. This linear compression might lose complex, non-linear feature interactions compared to models that learn to attend to all features directly.



---

---

> ### Author Rebuttal · Authors · 2026-03-31
>
> We sincerely thank the reviewer for the very positive assessment and for accurately capturing the paper’s main contributions, including the efficiency-accuracy trade-off of a row-wise PFN backbone, the per-query online early-exit design, and the unified treatment of classification and regression. We are especially encouraged that the reviewer finds the work technically solid and practically meaningful. We address the reviewers' suggestions and concerns in a Q&A format:
>
> > Q1. On novelty
>
> We agree that gated attention, register tokens, and early exit are established ideas in other domains. Our contribution is not to claim these components as individually novel, but to show that, in **tabular in-context learning**, their careful integration is sufficient to make a lightweight **row-attention-only PFN** highly competitive again, while preserving simplicity, efficiency, and enabling **per-query adaptive inference**. We will clarify this positioning more explicitly in the final version.
>
> > Q2. The scaling term in Eq. (1)
>
> Thank you for pointing this out. We checked the submitted PDF, and Eq. (1) indeed uses the standard scaling factor $\sqrt{d_h}$. We suspect this may be due to a PDF rendering/display issue on the review side. We will make sure the notation is clearly rendered and unambiguous in the final version.
>
> > Q3. Performance as the number of instances (N) grows beyond 20,000
>
> This is an important question. Since SwiftPFN uses **row-wise self-attention**, its cost is quadratic in the number of rows, so very large-(N) regimes will eventually become a bottleneck compared with methods explicitly designed for larger-sample settings.
>
> Our current work mainly targets the **small-to-medium tabular prediction regime** represented by PFN-style in-context learning, and our continued pretraining therefore focuses on up to **20k rows**. Following the reviewer’s suggestion, we additionally examined TALENT classification datasets with more than **20,000** training samples and computed the average rank across methods:
>
> |Method|Avg. Rank|
> |-|-|
> |RealMLP|9.31|
> |ModernNCA|9.69|
> |TabM|11.65|
> |**SwiftPFN**|**12.32**|
> |TabICL|12.56|
> |CatBoost|16.04|
> |LightGBM|16.10|
> |TabPFN v2|16.61|
>
> These results suggest that, once we move beyond the pretraining row range, **SwiftPFN does show a performance drop**, and non-TFM methods can become more favorable in this regime. We will make this limitation more explicit in the discussion and mention scaling to larger-(N) datasets as an important direction for future work.
>
> > Q4. High-feature datasets and PCA
>
> We agree that this is an important practical concern. PCA is only applied when the feature dimensionality exceeds the backbone budget, in which case we reduce the features to 100 dimensions. We additionally evaluated 9 high-dimensional datasets (2,420–19,993 features). Using the same ensemble protocol as in the main benchmark, SwiftPFN obtains an average rank of 2.1. We do not observe a clear monotonic degradation with increasing dimensionality, although SwiftPFN is overall less strong on such datasets than on standard benchmarks. The full per-dataset results are provided in our response (Q5) to Reviewer xfDg and will be included in the final version.
>
> > Q5. Comparison to TabICL v2
>
> Thank you for raising this interesting point. TabICL v2 is a post-deadline work that we have also followed closely, and we agree that it is a stronger recent TFM baseline on TALENT. Our focus here, however, is somewhat different: we study how far a **lightweight row-attention-only PFN backbone** can be pushed through architectural improvements while keeping **inference cost low**. As discussed in **Sec. 4.2**, our continued pretraining still uses the earlier **v1-style data engine**, while TabICL v2 leverages an improved data engine. Despite this, SwiftPFN already achieves competitive performance with a substantially lighter architecture. We believe this suggests that the row-attention-only route still has meaningful headroom, and that our proposed modifications are largely **orthogonal** to the advances used in newer systems such as TabICL v2. We will discuss this more explicitly in the final version.
>
> > Q6. Regression loss formulation
>
> Thank you for asking for clarification. Our training objective is:
>
> $$
> L_{CE} + 0.5 \times L_{MSE} + 0.5 \times L_{MAE}
> $$
>
> We use this simple equally weighted combination because, in practice, it yields a loss scale close to the stabilized scale of the classification loss during training, which helps prevent the unified pretraining process from being overly biased toward one task type. We also empirically found that using a mixed **MSE+MAE** objective performs slightly better than using either **MSE** or **MAE** alone. We will add this exact formulation and motivation to the paper for clarity.

---

> > ### Author Rebuttal · Reviewer_hqtJ · 2026-04-04
> >
> > Thank you for the clarifications and additional results. Will update my score accordingly.

---

> > > ### Author Response · Authors · 2026-04-04
> > >
> > > Thank you for the update. We are glad that the additional clarifications and results were helpful. We also appreciate your constructive comments throughout the review process, which helped improve the quality of our work.

---

### Official Review · Reviewer_xfDg · 2026-03-13

**Soundness:** 3
**Presentation:** 2
**Significance:** 3
**Originality:** 3
**Overall Recommendation:** 5
**Confidence:** 4

**Summary:**

This paper revisits the TabPFNv1 architecture for its lightweight nature stemming from its row-wise attention-only design. It introduces changes to make its performance comparable to state-of-the-art tabular foundation models while further improving its already fast inference time, the resulting model is called SwiftPFN.
The changes introduced in SwiftPFN are element-wise gated attention to stabilize pretraining, learnable registers that capture global dataset information, and an early-exiting mechanism that can choose to generate a prediction after passing the data through only a subset of the model layers.
The paper evaluates SwiftPFN on a large benchmark of tabular regression and classification tasks and includes ablations to study the contribution of the proposed components.

**Compliance With Llm Reviewing Policy:**

Affirmed.

**Final Justification:**

The author's rebuttal clarified several of my open questions and presented new results exploring possible limitations of the method. While some presentation improvements are left for the final version, I find this contribution important to the tabular foundation model community. In particular, the paper highlights that certain design decisions may have been prematurely dismissed and deserve further investigation.

I therefore upgrade my assessment to an "Accept".

**Key Questions For Authors:**

**Q1:**
It is unclear which model corresponds to label (1) in Figure 1. How does this model differ from TabPFN v1? The figure suggests that model (1) outperforms TabPFN v1 while using substantially fewer parameters. Do the authors have intuition for why this is the case?

**Q2:**
In Figure 4, it is unclear whether the models SPFN-S1, SPFN-S1-Gate, SPFN-S1-Register, SPFN-S1-Gate-Register, and SwiftPFN are evaluated with or without the early-exit mechanism. Could the authors clarify whether early exiting is enabled for these models in the reported results?

**Q3:**
The TabPFNv1 architecture is limited in the maximum number of features it can handle; if that limit is exceeded, SwiftPFN applies PCA for dimensionality reduction. This potentially limits SwiftPFN's performance on datasets with a large number of features. Do the authors have any insights into whether this is a practical concern? A figure showing comparative performance to other models given the number of features might strengthen the paper by addressing this concern.

**Limitations:**

yes

**Strengths And Weaknesses:**

**Soundness**

The paper is technically sound. It evaluates and compares SwiftPFN against an extensive set of baselines on a large benchmark of tabular regression and classification tasks, and includes ablations to study the contributions of the proposed components.
For the evaluation, both rank comparisons and PAMA values are considered; additionally, pairwise Wilcoxon signed-rank tests with Holm correction are performed to assess the statistical significance of performance differences between models.

However, when introducing the statistical tests, no citations were provided. Including the appropriate citations would improve methodological transparency.

**Presentation**

The choice of extending TabPFNv1 is well motivated by efficiency arguments, and the aThe decision to extend TabPFNv1 is well justified by efficiency considerations, and the architectural changes, including element-wise gated attention, learnable register tokens, and early-exit mechanisms, are easy to understand because the use of both figures and mathematical formulations helps clarify these components.

However, there are several issues with the organization and referencing in the experimental section. Figures 4 and 6 are not referenced in the corresponding parts of the text, which disrupts the flow of the presentation. In addition, the experimental setup underlying Figure 1 is not clearly discussed in the main text. Similarly, the main text does not reference Appendices C and D.

Finally, some implementation details are missing, which limits reproducibility. In particular, the PCA dimensionality used when the feature limit is exceeded and the number of register tokens employed by the model are not reported.

**Significance**

This paper revisits the TabPFNv1 architecture and demonstrates that, with relatively simple modifications, it can achieve performance comparable to state-of-the-art tabular foundation models. This finding is significant because it suggests that designs previously considered less competitive may still be promising with the right modifications. As such, the work could encourage further research into understanding which components of tabular foundation models contribute most to performance and how different architectural design choices interact.

**Originality**

The overall methodological novelty is somewhat limited. The proposed model builds on the existing TabPFN v1 architecture and incorporates several previously studied components, including element-wise gated attention, learnable register tokens, and early-exit mechanisms. While these individual ideas are not new in themselves, their application and combination in the context of tabular foundation models is novel. The main contribution, therefore, lies in demonstrating that these relatively simple modifications can substantially improve the performance–efficiency trade-off of the TabPFN architecture.

---

> ### Author Rebuttal · Authors · 2026-03-31
>
> We sincerely thank the reviewer for the careful reading and for recognizing the paper’s technical soundness, clear motivation, and the value of revisiting the lightweight TabPFNv1-style design. We are also glad that the reviewer found the proposed modifications easy to follow. We address the reviewers' suggestions and concerns in a Q&A format:
>
> > Q1 About novelty
>
> We agree that gated attention, register tokens, and early exit are not individually new in isolation. Our contribution is to show that, in tabular in-context learning, their careful integration can substantially improve a row-attention-only PFN backbone while preserving its simplicity and efficiency. In other words, our main message is that the original PFN v1 design space still has substantial room for further development, and that a lightweight row-wise-only architecture can remain highly competitive when paired with an improved pretraining recipe and a few carefully chosen modifications. We will clarify this positioning more explicitly in the final version.
>
> > Q2 On presentation and reproducibility
>
> Thank you for pointing out the presentation issues. We will improve the paper accordingly in the final version. In particular, we will:
> * add appropriate citations for the statistical tests used in the benchmark,
> * explicitly reference Figs. 4 and 6, as well as App. C/D, in the relevant parts of the main text,
> * clarify the experimental setup behind Fig. 1 in the caption and main text,
> * report the omitted implementation details, including the number of register tokens (32) and the PCA target dimensionality (100).
>
> > Q3: What does model (1) in Fig. 1 represent, and why can it outperform TabPFNv1 with fewer parameters?
>
> Model (1) is SwiftPFN without gated attention and without register tokens. It has fewer parameters than TabPFNv1 because it uses an embedding dimension of 192 (vs. 512 in TabPFNv1). Its better performance likely comes from an improved pretraining recipe, including more effective training and TabICL-style GBDT-based synthetic data generation. This supports our main claim that the original PFN v1 row-attention-only backbone still has substantial headroom.
>
> > Q4: Are early-exit versions included in Figs. 4 and 6?
>
> The rank comparison in Fig. 4 and the timing comparison in Fig. 6 exclude early-exit variants.
>
> Including many early-exit variants under different thresholds would introduce multiple highly similar SwiftPFN operating points. In rank-based comparisons, this could artificially enlarge the rank gaps to other methods and make the comparison less interpretable. We therefore evaluate early-exit variants separately across multiple thresholds in Fig. 7. In the final version, we will consider presenting them more systematically, for example by providing separate rank-style figures for representative thresholds.
>
> > Q5: Does PCA become a practical concern on high-dimensional datasets?
>
> We agree that this is an important practical concern. PCA is only applied when the feature dimension exceeds the fixed input budget of the PFN-style backbone, and in such cases we reduce the feature dimension to 100.
>
> Following the reviewer's suggestion, we additionally evaluated 9 high-dimensional datasets used in *A Closer Look at TabPFN v2: Understanding Its Strengths and Extending Its Capabilities*. We selected datasets with more than 150 samples to ensure robust evaluation. Following the evaluation protocol from the original paper, we used 5 train-test splits with 3 random seeds per split. We compared TabPFN v2, SwiftPFN, TabICL, and XGBoost. To remain consistent with our main benchmark protocol, SwiftPFN and TabICL use 16-model ensembles, while TabPFN v2 uses a 4-model ensemble, exactly as in our main experiments.
>
> The detailed results are shown below.
>
> |Dataset|#dim|TabPFN v2|SwiftPFN|TabICL|XGBoost|
> |-|-|-|-|-|-|
> |warpPIE10P|2420|1|1|1|0.95|
> |PCMAC|3289|0.93|0.92|0.89|0.92|
> |lung|3312|0.95|0.92|0.95|0.94|
> |RELATHE|4322|0.86|0.91|0.87|0.87|
> |BASEHOCK|4862|0.98|0.98|0.95|0.95|
> |gisette|5000|0.97|0.99|0.96|0.98|
> |TOX_171|5748|0.81|0.79|0.83|0.78|
> |arcene|10000|0.84|0.81|0.79|0.75|
> |SMK_CAN_187|19993|0.71|0.70|0.69|0.66|
> |Avg. Rank||1.8|2.1|2.9|3.2|
>
> Although SwiftPFN is overall somewhat weaker on such high-dimensional datasets than on standard tabular benchmarks, we do not observe a clear monotonic degradation trend as the feature dimensionality increases. We note that high-dimensional datasets remain challenging for current TFM-based methods, and PCA-based dimensionality reduction appears to be a relatively strong and practical solution in this regime. We will include this additional table in the final version and discuss the limitations of SwiftPFN on very high-dimensional problems more explicitly in the discussion/limitations section.

---

> > ### Author Rebuttal · Reviewer_xfDg · 2026-04-03
> >
> > The author's rebuttal clarified several of my open questions and presented new results exploring possible limitations of the method. While some presentation improvements are left for the final version, I find this contribution important to the tabular foundation model community. In particular, the paper highlights that certain design decisions may have been prematurely dismissed and deserve further investigation.
> >
> > I therefore plan to upgrade my assessment to an "Accept".

---

> > > ### Author Response · Authors · 2026-04-03
> > >
> > > Thank you very much for the positive update. We are glad that our rebuttal helped clarify your concerns and that you found the contribution valuable to the tabular foundation model community. We also sincerely appreciate your constructive feedback, which has helped us improve the quality and presentation of the paper. We will incorporate the remaining presentation improvements in the final version.

---

### Decision · Program_Chairs · 2026-04-30

**Decision:**

Accept (spotlight)

**Comment:**

There is consensus among reviewers that this is a high-quality paper that should be included in the ICML program. The efficiency of row-based approaches has been relatively understudied as the field has mostly moved to cell-based approaches. To my knowledge, the early exit mechanism is novel for TFMs. The results showing high efficiency are strong; these results are important as TFMs begin to scale to larger and larger data.

I assume that the authors will make the requested presentation changes in time for the camera-ready submission.